# Metabolic clogging of mannose triggers dNTP loss and genomic instability in human cancer cells

Yoichiro Harada[1]\*, Yu Mizote[2], Takehiro Suzuki[3], Akiyoshi Hirayama[4,5], Satsuki Ikeda[4], Mikako Nishida[6], Toru Hiratsuka[7], Ayaka Ueda[8], Yusuke Imagawa[7], Kento Maeda[1], Yuki Ohkawa[1], Junko Murai[4,9,10], Hudson H Freeze[11], Eiji Miyoshi[8], Shigeki Higashiyama[7,9,10], Heiichiro Udono[6], Naoshi Dohmae[3], Hideaki Tahara[2,12], Naoyuki Taniguchi[1]

[1]Department of Glyco-Oncology and Medical Biochemistry, Research Institute, Osaka International Cancer Institute, Osaka, Japan; [2]Department of Cancer Drug Discovery and Development, Research Institute, Osaka International Cancer Institute, Osaka, Japan; [3]Biomolecular Characterization Unit, RIKEN Center for Sustainable Resource Science, Saitama, Japan; [4]Institute for Advanced Biosciences, Keio University, Yamagata, Japan; [5]Systems Biology Program, Graduate School of Media and Governance, Keio University, Kanagawa, Japan; [6]Department of Immunology, Okayama University Graduate School of Medicine, Dentistry and Pharmaceutical Sciences, Okayama, Japan; [7]Department of Oncogenesis and Growth Regulation, Research Institute, Osaka International Cancer Institute, Osaka, Japan; [8]Department of Molecular Biochemistry and Clinical Investigation, Graduate School of Medicine, Osaka University, Osaka, Japan; [9]Division of Cell Growth and Tumor Regulation, Proteo-Science Center, Ehime University, Ehime, Japan; [10]Department of Biochemistry and Molecular Genetics, Graduate School of Medicine, Ehime University, Ehime, Japan; [11]Human Genetics Program, Sanford Burnham Prebys Medical Discovery Institute, La Jolla, United States; [12]Project Division of Cancer Biomolecular Therapy, Institute of Medical Science, The University of Tokyo, Tokyo, Japan

\*For correspondence:
yoharada3@oici.jp

Competing interest: The authors declare that no competing interests exist.

**Abstract** Mannose has anticancer activity that inhibits cell proliferation and enhances the efficacy of chemotherapy. How mannose exerts its anticancer activity, however, remains poorly understood. Here, using genetically engineered human cancer cells that permit the precise control of mannose metabolic flux, we demonstrate that the large influx of mannose exceeding its metabolic capacity induced metabolic remodeling, leading to the generation of slow-cycling cells with limited deoxy-ribonucleoside triphosphates (dNTPs). This metabolic remodeling impaired dormant origin firing required to rescue stalled forks by cisplatin, thus exacerbating replication stress. Importantly, pharmacological inhibition of de novo dNTP biosynthesis was sufficient to retard cell cycle progression, sensitize cells to cisplatin, and inhibit dormant origin firing, suggesting dNTP loss-induced genomic instability as a central mechanism for the anticancer activity of mannose.

## Editor's evaluation

Mannose is toxic to honeybees and some cancer cells that lack sufficient capacity to metabolize this sugar. However, the precise metabolic consequences of impaired mannose metabolism require

further understanding. In this important study, Harada et al. provide convincing evidence that an inability to metabolize mannose leads to impaired synthesis of deoxynucleotide triphosphates and DNA, which can be leveraged to sensitize cancer cells to chemotherapy.

## Introduction

In mammals, mannose is a monosaccharide that is essential for life and is synthesized de novo from glucose through a glycolysis branch (*Figure 1—figure supplement 1A*). This process requires the action of mannose phosphate isomerase (MPI), the enzyme that catalyzes the interconversion between fructose-6-phosphate (Fruc-6-P) and mannose-6-phosphate (Man-6-P) (*Alton et al., 1998*). Man-6-P is further converted to GDP-mannose for the biosynthesis of asparagine-linked glycans (N-glycans) in the endoplasmic reticulum (*Harada et al., 2013*). In the secretory pathway, extensive trimming of N-glycans by mannosidases generates free mannose, which is secreted from cells and contributes to the extracellular pool of mannose (30–130 µM in blood) (*Sharma and Freeze, 2011*). The extracellular mannose can be taken up by cells and the salvaged mannose is directly converted to Man-6-P, the majority of which is efficiently directed into glycolysis by the action of MPI for unknown reasons (*Ichikawa et al., 2014*; *Sharma and Freeze, 2011*).

The large influx of mannose is known to suppress cell proliferation and enhance the efficacy of chemotherapy, particularly in cancer cells that express low levels of MPI (*Gonzalez et al., 2018*), although the underlying mechanisms remain poorly understood. It has been known for nearly a century that feeding mannose to honeybees, which are believed to express negligible amounts of MPI (*Sols et al., 1960*), is lethal to these insects (*Staudenmayer, 1939*). This is because the intracellular levels of Man-6-P exceed the capacity to metabolize it, and the excess Man-6-P inhibits glucose metabolism and decreases the intracellular ATP pool (*DeRossi et al., 2006*; *Sols et al., 1960*). This metabolic deficiency is called honeybee syndrome; however, it is unknown whether this syndrome plays a key role in the anticancer activity of mannose, and if so, what metabolic checkpoints are targeted by mannose to trigger its anticancer activity. Moreover, it is enigmatic how mannose sensitizes poorly proliferating cancer cells to chemotherapy that is designed to target actively proliferating cells. In this study, we addressed these two fundamental questions by using *MPI*-knockout (MPI-KO) human cancer cells as a model system for honeybee syndrome.

## Results
### Induction of honeybee syndrome suppresses cell proliferation and increases chemosensitivity

To establish MPI-KO human cancer cells using the CRISPR–Cas9 system, we exploited the mannose auxotrophy and sensitivity observed in MPI-KO mouse embryonic fibroblasts (MPI-KO MEFs) (*DeRossi et al., 2006*). The addition of a physiological concentration of mannose (50 µM, unchallenged) to culture medium supported the proliferation of MPI-KO MEFs (*Figure 1—figure supplement 1B*). In contrast, mannose starvation or the addition of a supraphysiological concentration of mannose (5 mM, challenged) suppressed the proliferation of MPI-KO MEFs, but not that of wild-type MEFs (*Figure 1—figure supplement 1C*). On these bases, we knocked out the *MPI* gene in human fibrosarcoma HT1080 cells and screened the gene-edited clones under mannose-unchallenged conditions. Three MPI-KO HT1080 clones were obtained (*Figure 1A*). In one clone (#1), cell division stopped under mannose-unchallenged conditions, but the other two clones (#2 and #3) could proliferate. These two clones exhibited mannose auxotrophy and sensitivity as expected (*Figure 1B* and *Figure 1—figure supplement 2A and B*), while the parental HT1080 cells showed marginal defects in cell proliferation at mannose concentrations higher than 15 mM (*Figure 1C*). The effects of mannose starvation and mannose challenge on the proliferation of MPI-KO cells were almost fully rescued by reintroduction of the human *MPI* gene (*Figure 1D–F* and *Figure 1—figure supplement 2C–E*), ruling out the potential off-target effects of gene editing. As expected, mannose challenge to MPI-KO HT1080 cells caused the dramatic accumulation of hexose-6-phosphate (the sum of glucose-6-phosphate and Man-6-P) compared with that under mannose-unchallenged conditions (*Figure 1G*). Consistent with the essential role of exogenous mannose in the production of Man-6-P and therefore that of GDP-mannose for

**eLife digest** In order to grow and divide, cells require a variety of sugars. Breaking down sugars provides energy for cells to proliferate and allows them to make more complex molecules, such as DNA. Although this principle also applies to cancer cells, a specific sugar called mannose not only inhibits cancer cell division but also makes them more sensitive to chemotherapy. These anticancer effects of mannose are particularly strong in cells lacking a protein known as MPI, which breaks down mannose.

Evidence from honeybees suggests that a combination of mannose and low levels of MPI leads to a build-up of a modified form of mannose, called mannose-6-phosphate, within cells. As a result, pathways required to release energy from glucose become disrupted, proving lethal to these insects. However, it was not clear whether the same processes were responsible for the anticancer effects of mannose.

To investigate, Harada et al. removed the gene that encodes the MPI protein in two types of human cancer cells. The experiments showed that mannose treatment was not lethal to these cells but overall slowed the cell cycle – a fundamental process for cell growth and division. More detailed biochemical experiments showed that cancer cells with excess mannose-6-phosphate could not produce the molecules required to make DNA. This prevented them from doubling their DNA – a necessary step for cell division – and responding to stress caused by chemotherapy.

Harada et al. also noticed that cancer cells lacking MPI did not all react to mannose treatment in exactly the same way. Therefore, future work will address these diverse reactions, potentially providing an opportunity to use the mannose pathway to search for new cancer treatments.

N-glycan biosynthesis in MPI-KO MEFs (*Harada et al., 2013*), mannose starvation severely decreased N-glycosylation in MPI-KO HT1080 cells (*Figure 1H*). However, mannose challenge showed negligible effects on N-glycosylation, indicating that mannose challenge suppresses cell proliferation through a mechanism distinct from N-glycosylation defects.

Mannose challenge increased the sensitivity of MPI-KO HT1080 cells to DNA replication inhibitors (i.e., cisplatin and doxorubicin) when the cells had been preconditioned with excess mannose prior to the drug treatment (*Figure 1I and J* and *Figure 1—figure supplement 2F and G*). We also generated MPI-KO HeLa cells as another cell model (*Figure 1—figure supplement 3A–I*) and found that mannose challenge also increased the sensitivity of these cells to cisplatin and doxorubicin (*Figure 1—figure supplement 3J and K*). All of these results demonstrate that the induction of honeybee syndrome suppresses cell proliferation and increases chemosensitivity in our MPI-KO human cancer cell models. We mostly used MPI-KO HT1080 (#3) cells for subsequent study, while the other cell models showed similar results.

## Mannose challenge generates slow-cycling cells

Cell proliferation is tightly controlled by cell cycle progression. To explore the mechanism behind the antiproliferative activity of mannose, we compared the cell cycle progression between mannose-challenged and -unchallenged MPI-KO HT1080 cells by using a two-color fluorescent ubiquitination-based cell cycle indicator [Fucci(CA)] (*Sakaue-Sawano et al., 2017*). In this reporter system, $G_1$ phase was defined by the exclusive expression of mCherry-hCdt1(1/100)Cy(−), which was rapidly turned off upon the onset of S phase where mVenus-hGem(1/110) gradually accumulated (*Figure 2A and B* and *Video 1*). The mCherry-hCdt1(1/100)Cy(−) was re-expressed upon the onset of $G_2$ phase (the double-positive phase), followed by the termination of M phase where mVenus-hGem(1/110) was abruptly turned off. Under mannose-unchallenged conditions, MPI-KO HT1080 cells showed exponential growth (*Figure 2C*) and a typical Fucci(CA) signal profile (*Figure 2B and D*), as reported in HeLa cells (*Sakaue-Sawano et al., 2017*). In contrast, mannose challenge almost completely suppressed cell proliferation (*Figure 2C*) and significantly prolonged the cell cycle, showing a variety of atypical Fucci(CA) signal profiles (*Figure 2E and F* and *Figure 2—figure supplement 1A*). We classified these profiles based on the order of expression of Fucci(CA) reporters [mCherry-hCdt1(1/100)Cy(−) as R, mVenus-hGem(1/110) as G, and the double-negative phase as Dn, *Figure 2—figure supplement 1A*]. The classification analysis revealed that a small proportion of the mannose-challenged cells showed

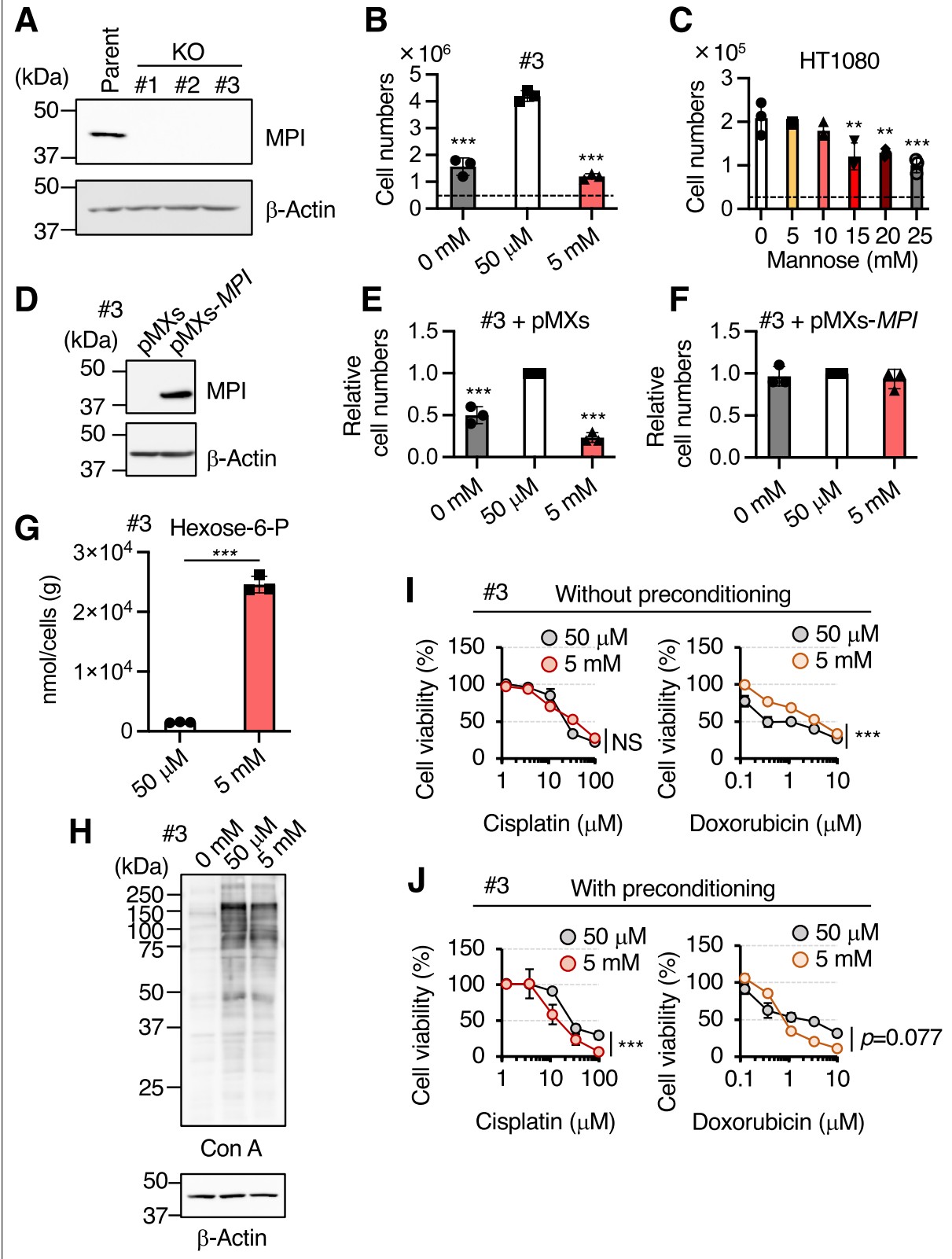

**Figure 1.** The induction of honeybee syndrome suppresses cell proliferation and increases chemosensitivity. (**A**) Western blot analysis of HT1080 (parent) and mannose phosphate isomerase knockout (MPI-KO) HT1080 (KO, clone #1–3) cells. The blots have been vertically flipped for presentation purpose. (**B, C**) The cell numbers of MPI-KO HT1080 (#3, **B**) and the parental HT1080 (**C**) cells after 48 hr incubation in culture medium supplemented with mannose at the indicated concentrations. A dashed line indicates the cell numbers at seeding. (**D**) Western blot analysis of MPI-KO HT1080 (#3)

*Figure 1 continued*

cells retrovirally transduced with empty vector (pMXs) or human *MPI* gene (pMXs-*MPI*). (**E, F**) The relative cell numbers of MPI-KO HT1080 (#3) cells retrovirally transduced with empty vector (pMXs, **E**) or human *MPI* gene (pMXs-*MPI*, **F**) after 48 hr incubation under mannose-starved (0 mM), mannose-unchallenged (50 µM), or mannose-challenged (5 mM) conditions. (**G**) The quantification of hexose-6-P in MPI-KO HT1080 (#3) cells cultured in the presence of 50 µM or 5 mM mannose. (**H**) Lectin blot and western blot analyses of MPI-KO HT1080 (#3) cells cultured as in (**B**). Con A, concanavalin A lectin. (**I**) Cell viability assay in MPI-KO HT1080 (#3) cells co-treated with mannose (50 µM or 5 mM) and DNA replication inhibitors (cisplatin or doxorubicin) for 24 hr (**I**, without preconditioning), or preconditioned with mannose (50 µM or 5 mM) for 24 hr, followed by incubation with the DNA replication inhibitors for an additional 24 hr in the presence of the same concentrations of mannose used for preconditioning (**J**, with preconditioning). Data represent the mean ± SD; n = 3 independent experiments. *p<0.05, **p<0.01, and ***p<0.001, NS, not significant, one-way ANOVA with *post hoc* Dunnett's test (**B**, **C**, **E**, **F**) Welch's *t*-test (**G**), or two-way ANOVA with *post hoc* Bonferroni's test (**I**, **J**).

The online version of this article includes the following source data and figure supplement(s) for figure 1:

**Source data 1.** Original blot images depicting cropped regions for *Figure 1A*.

**Source data 2.** Original blot images depicting cropped regions for *Figure 1D*.

**Source data 3.** Original blot images depicting cropped regions for *Figure 1H*.

**Figure supplement 1.** Mannose auxotrophy and sensitivity in mannose phosphate isomerase knockout (MPI-KO) mouse embryonic fibroblasts (MEFs) and wild-type MEFs.

**Figure supplement 2.** Establishment of mannose phosphate isomerase knockout (MPI-KO) HT1080 cells.

**Figure supplement 2—source data 1.** Original blot images depicting cropped regions for *Figure 1—figure supplement 2C*.

**Figure supplement 3.** Establishment of mannose phosphate isomerase knockout (MPI-KO) HeLa cells.

**Figure supplement 3—source data 1.** Original blot images depicting cropped regions for *Figure 1—figure supplement 3A*.

**Figure supplement 3—source data 2.** Original blot images depicting cropped regions for *Figure 1—figure supplement 3D*.

**Figure supplement 3—source data 3.** Original blot images depicting cropped regions for *Figure 1—figure supplement 3E*.

normal-like but strikingly extended Fucci(CA) signal profiles (*Figure 2E and F* and *Figure 2—figure supplement 1A*, normal-like, 12% of the total). Notably, these normal-like cell populations frequently failed to undergo cytokinesis in M phase and re-entered $G_1$ phase without generating two equivalent daughter cells (*Video 2*). Other fractions included cells that were arrested in $G_1$ phase (*Figure 2E and F* and *Figure 2—figure supplement 1A*, RRR, 26.7%), that showed little to no double-positive phase (i.e., the $G_2$-to-M phase) (*Figure 2E and F* and *Figure 2—figure supplement 1A*, RGR, 15.8% and GRG, 9.9%, GDn, 5.9%, RGG, 5.0%, RGDn, 4.0%; 40.6% of total), or that remained in the double-negative phase (*Figure 2F* and *Figure 2—figure supplement 1A*, Dn, 8.9%), which is normally seen only at the $G_1$/S transition (*Sakaue-Sawano et al., 2017*). These results suggest that mannose challenge severely impairs the entry of the cells into S phase and its progression to mitotic phase. Strikingly, however, switching of the mannose-challenge medium to the mannose-unchallenged medium after long-term mannose challenge (6 d) resulted in robust cell proliferation (*Figure 2G*) with cell cycle progression indistinguishable from that of mannose-unchallenged cells (*Figure 2—figure supplement 2A–D*), suggesting that some fraction of the mannose-challenged cells had entered a quiescent state upon mannose challenge. Supporting this assumption, mannose challenge induced early accumulation of the cell cycle inhibitors p21 and p27 (*Abukhdeir and Park, 2008*; *Figure 2H*). To directly measure DNA synthesis, mannose-challenged and -unchallenged cells were pulse-labeled with 5-bromo-2'-deoxyuridine (BrdU) and stained with Hoechst 33342 for DNA. BrdU was incorporated into DNA in both mannose-challenged and -unchallenged cells, but mannose challenge gradually decreased cell populations that were actively synthesizing DNA over the incubation time (*Figure 2I and J*). Small proportions of the mannose-challenged cells exhibited DNA content greater than 4n after 2-day mannose challenge, which also exhibited BrdU incorporation, suggesting that these cell populations underwent endoreplication (*Edgar and Orr-Weaver, 2001*). Collectively, these results indicate that mannose challenge suppresses cell proliferation through a complex mechanism involving extremely slow cell cycle progression.

## Mannose challenge limits DNA synthesis at ongoing replication forks

We adopted a proteomic approach to dissect the molecular mechanism by which mannose challenge generated slow-cycling cells in MPI-KO HT1080 cells. Of over 7000 proteins identified in the proteomic datasets (*Source data 1*), proteins that were significantly up- and downregulated relative to those in the cells cultured for 1 d under mannose-unchallenged conditions were extracted at each

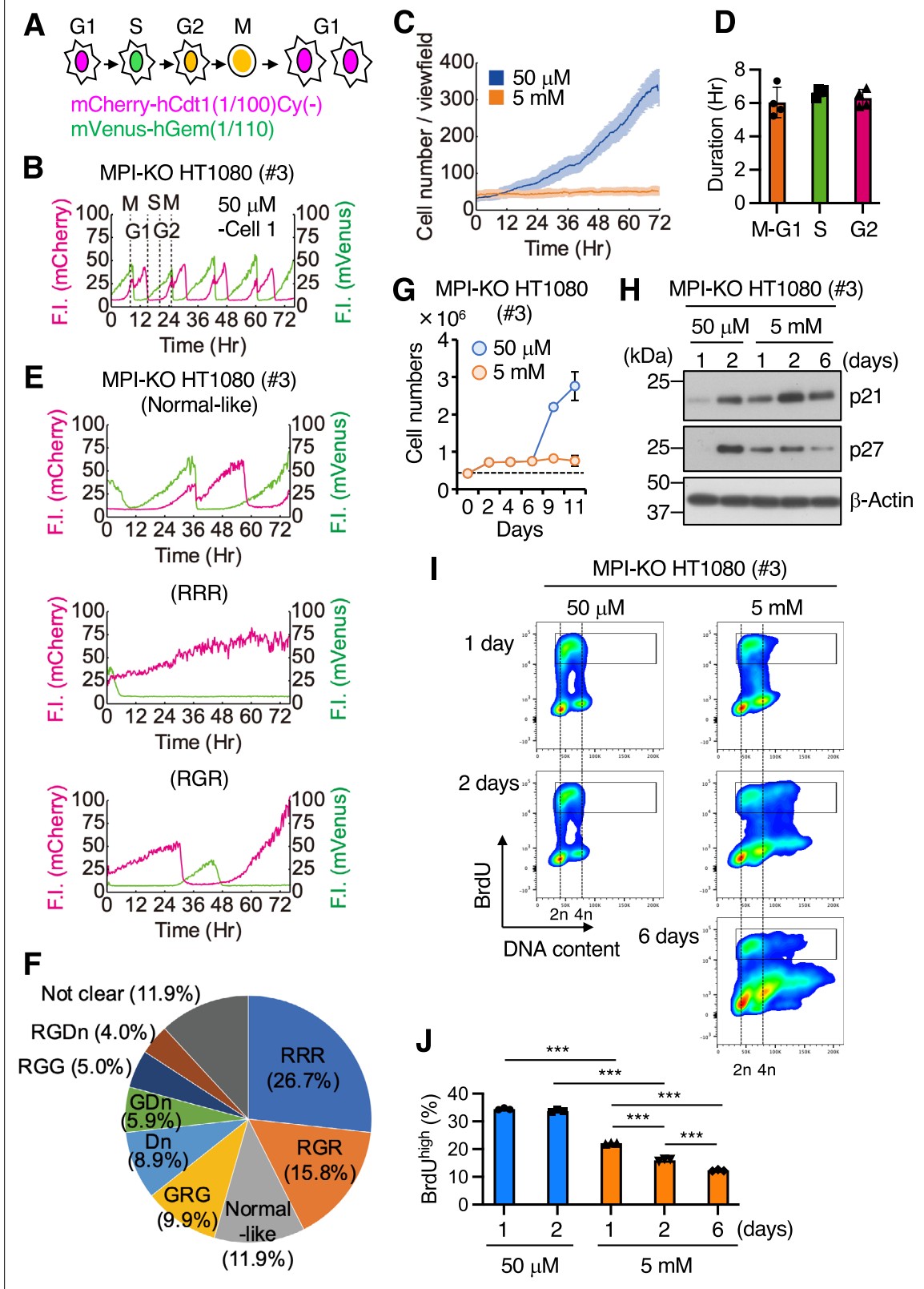

**Figure 2.** Mannose challenge generates slow-cycling cells. (**A**) Schematic representation of cell cycle progression (G1, S, G2, and M phases) visualized by Fucci(CA). Magenta, green, and yellow indicate the expression of mCherry-hCdt1(1/100)Cy(−), mVenus-hGem(1/110), and both, respectively. (**B**) A representative Fucci(CA) signal profile in mannose phosphate isomerase knockout (MPI-KO) HT1080 cells under the mannose-unchallenged conditions (50 μM). Fluorescent intensity (F.I.) for mCherry (left y-axis) and mVenus (right y-axis) is shown by magenta and green, respectively. Individual phases

*Figure 2 continued on next page*

*Figure 2 continued*

of the cell cycle (G$_1$, S, G$_2$, and M) are demarcated by dashed lines. (**C**) The number of MPI-KO HT1080 cells during the time-lapse imaging under mannose-unchallenged (50 µM, blue) and mannose-challenged (5 mM, orange) culture conditions. (**D**) The durations of M–G$_1$ (the sum of both phases), S, and G$_2$ phases (n=1109 cells) in MPI-KO HT1080 cells under mannose-unchallenged conditions (50 µM). (**E**) Representative Fucci(CA) signal profiles of MPI-KO HT1080 cells cultured under mannose-challenged conditions for 76 hr. The order of expression of mCherry-hCdt1(1/100)Cy(−) (denoted by R) and mVenus- hGem(1/110) (denoted by G) was used to classify the Fucci profiles and the classification is indicated in parentheses. See also *Figure 2—figure supplement 1*. (**F**) The proportion of Fucci(CA) signal profiles of the mannose-challenged MPI-KO HT1080 cells. R, mCherry-hCdt1(1/100)Cy(−) positive; G, mVenus-hGem(1/100) positive; Dn, double negative for Fucci indicators. One hundred cells were visually inspected for the classification and the cells that could not be classified were categorized as 'not clear.' See also *Figure 2—figure supplement 1*. (**G**) The number of MPI-KO HT1080 (#3) cells. The cells were cultured in the presence of 5 mM mannose for 6 d, and they were further cultured in the presence of 50 µM or 5 mM mannose for 5 d. A dashed line indicates the cell numbers at seeding. (**H**) Western blot analysis of MPI-KO HT1080 (#3) cells cultured in the presence of 50 µM or 5 mM mannose for the indicated time. (**I**) Flow cytometry for BrdU and DNA content (Hoechst33342) in MPI-KO HT1080 (#3) cells cultured in the presence of 50 µM or 5 mM mannose for the indicated time. The cells were labeled with 10 µM BrdU for 1 hr before harvest. The highly replicating cells (BrdU[high]) were annotated with black boxes. (**J**) The percentage of BrdU[high] cells in (**I**) (in the gated populations). Total numbers of single cells were set to 100%. Data represent the mean ± SD; n = 4 independent fields (**C**) and n = 3 independent experiments (**G, J**). ***p<0.001, one-way ANOVA with *post hoc* Tukey's test (**J**).

The online version of this article includes the following source data and figure supplement(s) for figure 2:

**Source data 1.** Original blot images depicting cropped regions for *Figure 2H*.

**Figure supplement 1.** The classification of Fucci profiles in mannose phosphate isomerase knockout (MPI-KO) HT1080 cells cultured under mannose-challenged conditions.

**Figure supplement 2.** Time-lapse imaging of mannose phosphate isomerase knockout (MPI-KO) HT1080 cells.

time point (*Source data 2*). Functional annotation analysis of the proteomic data using DAVID bioinformatic resources (https://david.ncifcrf.gov) revealed the downregulation of proteins related to the cell cycle and DNA replication in mannose-challenged cells (*Figure 3A*), which were enriched with the mini-chromosome maintenance 2-7 (MCM2-7) complex (*Figure 3B*). Western blot analysis and quantitative polymerase chain reaction (qPCR) confirmed the decrease in the expression levels of MCM2-7 proteins (*Figure 3C*) and their genes (*Figure 3D*) during mannose challenge over 6 d. The MCM2-7 complex is a core component of DNA helicase that unwinds the DNA duplex at replication forks in S phase (*Jones et al., 2021*; *Rzechorzek et al., 2020*), and the complex also plays a central role in the licensing of replication origins in G$_1$ phase by forming a pre-replicative complex with the six-subunit origin recognition complex (ORC1-6), CDC6, and CDT1 on chromatin (*Frigola et al., 2017*; *Remus et al., 2009*; *Zhai et al., 2017*). CDC6 and CDT1, which were not detected in our proteomic analysis, were depleted after 2-day mannose challenge, whereas the ORC2 subunit was expressed at relatively constant levels (*Figure 3C*). These results indicate that mannose challenge induces slow proteomic alterations in origin licensing factors, while this cannot be an essential trigger for the generation of slow-cycling cells as these cells appeared immediately after mannose challenge.

We performed chromatin flow cytometry for MCM2 and DNA content to elucidate the impact

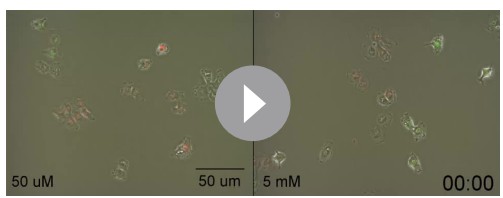

**Video 1.** Time-lapse imaging of cell cycles in mannose phosphate isomerase knockout (MPI-KO) HT1080 cells. MPI-KO HT1080 cells that expressed Fucci(CA) reporters were observed for 76 hr under the mannose-unchallenged (left) or the mannose-challenged (right) culture conditions. Red, mCherry-hCdt1(1/100)Cy(-); green, mVenus-hGem(1/110). Images were acquired every 15 min. Image size: 253.44 µm × 190.48 µm for each panel.

https://elifesciences.org/articles/83870/figures#video1

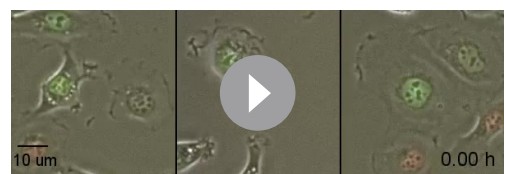

**Video 2.** Abnormal cytokinesis during M phase in the mannose-challenged mannose phosphate isomerase knockout (MPI-KO) HT1080 cells with normal-like Fucci(CA) signal profiles. Time-lapse movies of three representative Fucci(CA)-expressing MPI-KO HT1080 cells that were cultured under the mannose challenge conditions. Red, mCherry-hCdt1(1/100)Cy(-); green, mVenus-hGem(1/110). Note that the cells in G2-M phase (yellow) show abnormal cytokinesis. Images were acquired every 15 min. Image size: 52.8 µm × 52.8 µm for each panel.

https://elifesciences.org/articles/83870/figures#video2

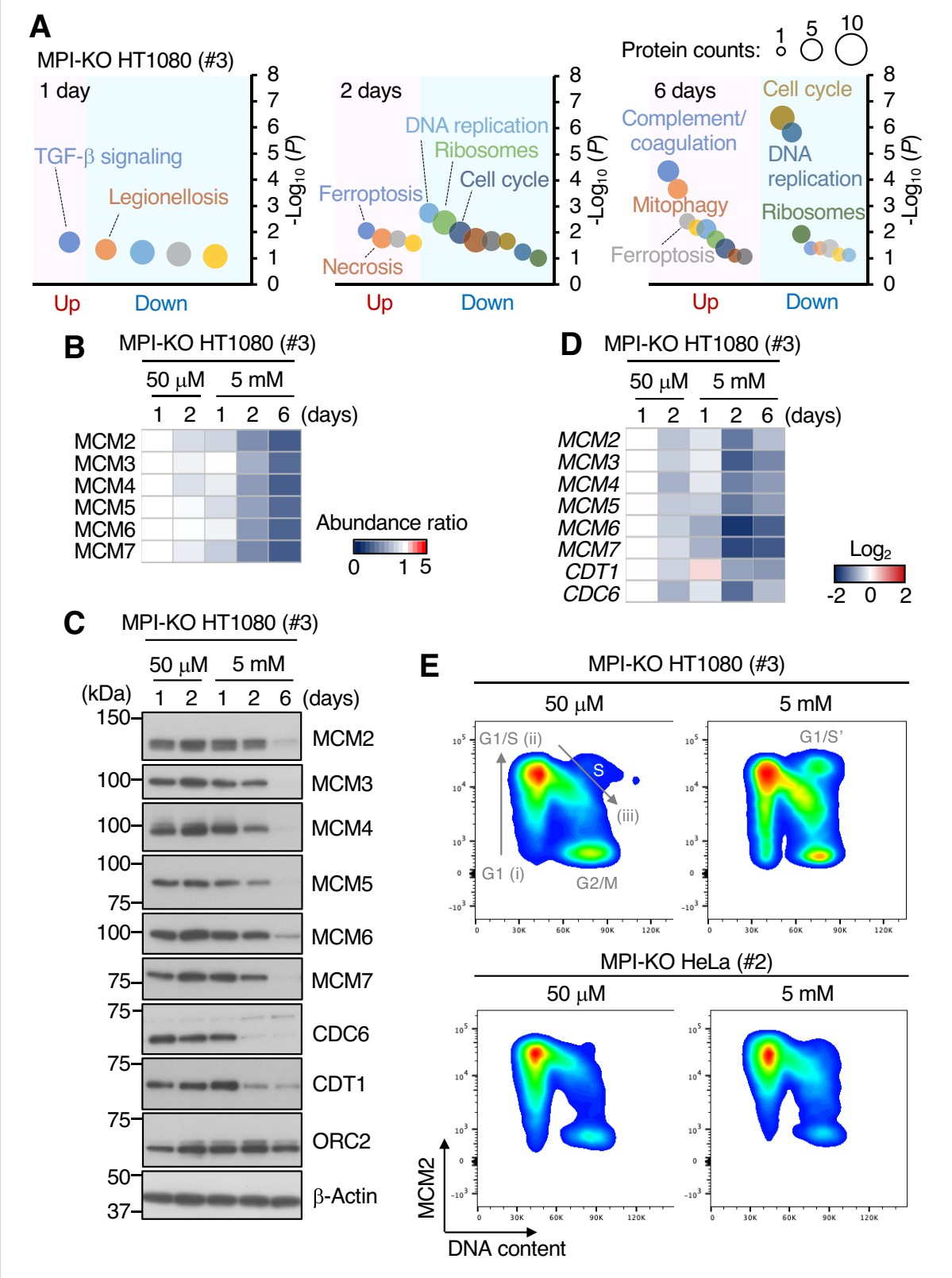

**Figure 3.** Mannose challenge limits DNA synthesis at ongoing replication forks. (**A**) Functional annotation analysis of the proteomic data in mannose phosphate isomerase knockout (MPI-KO) HT1080 (#3) cells cultured in the presence of 50 µM or 5 mM mannose for the indicated time. (**B**) Heatmap representation of the relative amounts of MCM2-7 proteins in MPI-KO HT1080 (#3) cells cultured in the presence of 50 µM or 5 mM mannose for the indicated time. (**C**) Western blot analysis of whole-cell lysates of MPI-KO HT1080 (#3) cells cultured in the presence of 50 µM or 5 mM mannose for

*Figure 3 continued on next page*

Figure 3 continued

the indicated time. (**D**) Heatmap representation of the relative expression levels of *MCM2*, *MCM3*, *MCM4*, *MCM5*, *MCM6*, *MCM7*, *CDT1*, and *CDC6* genes in MPI-KO HT1080 (#3) cells cultured in the presence of 50 μM or 5 mM mannose for the indicated time. (**E**) Chromatin flow cytometry for MCM2 and DNA content (Hoechst33342) in MPI-KO HT1080 (#3) or MPI-KO HeLa (#2) cells cultured in the presence of 50 μM or 5 mM mannose for 24 hr. The binding of MCM2 to chromatins (origin licensing) is detected in G1 phase (step i) and it peaks at G1/S boundary (step ii). As S phase progresses, MCM2 is dissociated from chromatins (step iii). $G_1/S'$ denotes the $G_1/S$ boundary of cell populations that underwent endoreplication.

The online version of this article includes the following source data for figure 3:

**Source data 1.** Original blot images depicting cropped regions for *Figure 3C*.

**Source data 2.** Original blot images depicting cropped regions for *Figure 3C*.

**Source data 3.** Original blot images depicting cropped regions for *Figure 3C*.

**Source data 4.** Original blot images depicting cropped regions for *Figure 3C*.

of mannose challenge on the chromatin-bound states of the MCM complex at the cell cycle level. Under the mannose-unchallenged conditions, the binding of MCM2 to chromatin took place in G1 phase (*Figure 3E*, step i) and peaked at the $G_1/S$ boundary (*Figure 3E*, step ii) in MPI-KO HT1080 cells and MPI-KO HeLa cells, indicating that replication origins were fully licensed (*Matson et al., 2017*). The cells with the fully licensed origins entered S phase as a tight population, and MCM2 dissociated from chromatin as two replication forks converged and terminated (*Figure 3E*, step iii; *Low et al., 2020*). Mannose challenge did not severely impair origin licensing (*Figure 3E*, step i), but the same treatment caused the accumulation of MCM2-positive chromatin in S phase (*Figure 3E*, step iii). Notably, the mannose-challenged cells were not actively incorporating BrdU into DNA (*Figure 2I and J* and *Figure 4—figure supplement 1A*), suggesting that ongoing replication forks were stuck on chromatin with little DNA synthesis under the mannose-challenged conditions. Taking these findings together, the limited DNA synthesis at ongoing replication forks likely contributes to the abnormally extended progression of S phase in the mannose-challenged cells.

## Mannose challenge disengages dormant origins from DNA synthesis during replication stress

Although our findings indicated that mannose challenge limits DNA synthesis at ongoing replication forks, its relevance to the increased chemosensitivity was still unclear. In humans, replication origins are licensed far more than actually used for DNA replication (*Langley et al., 2016*). The excess origins remain dormant under physiological conditions, while the dormant origins are activated for DNA replication when nearby replication forks stall upon encountering DNA lesions, thus preventing cells from the permanent replication arrest that leads to cell death (*Blow et al., 2011*; *Ge et al., 2007*; *Kawabata et al., 2011*; *Shima et al., 2007*). To test whether mannose challenge may also impair DNA synthesis from dormant origins during replication stress, we compared BrdU incorporation between the cells that were first pulsed with cisplatin to induce replication stress, followed by being left untreated or being treated with an inhibitor of the ataxia telangiectasia and Rad3-related protein (ATR) to forcibly activate dormant origins (*Moiseeva and Bakkenist, 2019*; *Moiseeva et al., 2019*; *Figure 4A*). In the mannose-unchallenged MPI-KO HT1080 cells (*Figure 4B and C*) and MPI-KO HeLa cells (*Figure 4—figure supplement 1A and B*), the cisplatin treatment alone partially suppressed the incorporation of BrdU, which was greatly recovered by ATR inhibition with VE-821 (ATRi) (*Charrier et al., 2011*; *Prevo et al., 2012*; *Reaper et al., 2011*), indicating that dormant origins are present in excess and their activation can engage in DNA synthesis during replication stress. In the mannose-challenged cells, however, the cisplatin treatment more severely reduced BrdU incorporation, which was barely restored by ATRi treatment (*Figure 4B and C* and *Figure 4—figure supplement 1A and B*). Forced activation of dormant origins during replication stress is known to cause the uncoupling of DNA unwinding and synthesis (*Murai et al., 2018*), leading to the accumulation of single-stranded DNA on chromatin marked by phosphorylation of the replication protein A2 (RPA2) (*Figure 4D*), confirming the activation of dormant origins by ATRi treatment in both mannose-challenged and -unchallenged cells. These results indicate that mannose challenge limits DNA synthesis from dormant origins during replication stress.

The replication initiation step requires chromatin loading of cell division cycle 45 (CDC45), one of the essential and limiting factors for replication initiation (*Moyer et al., 2006*), while this process

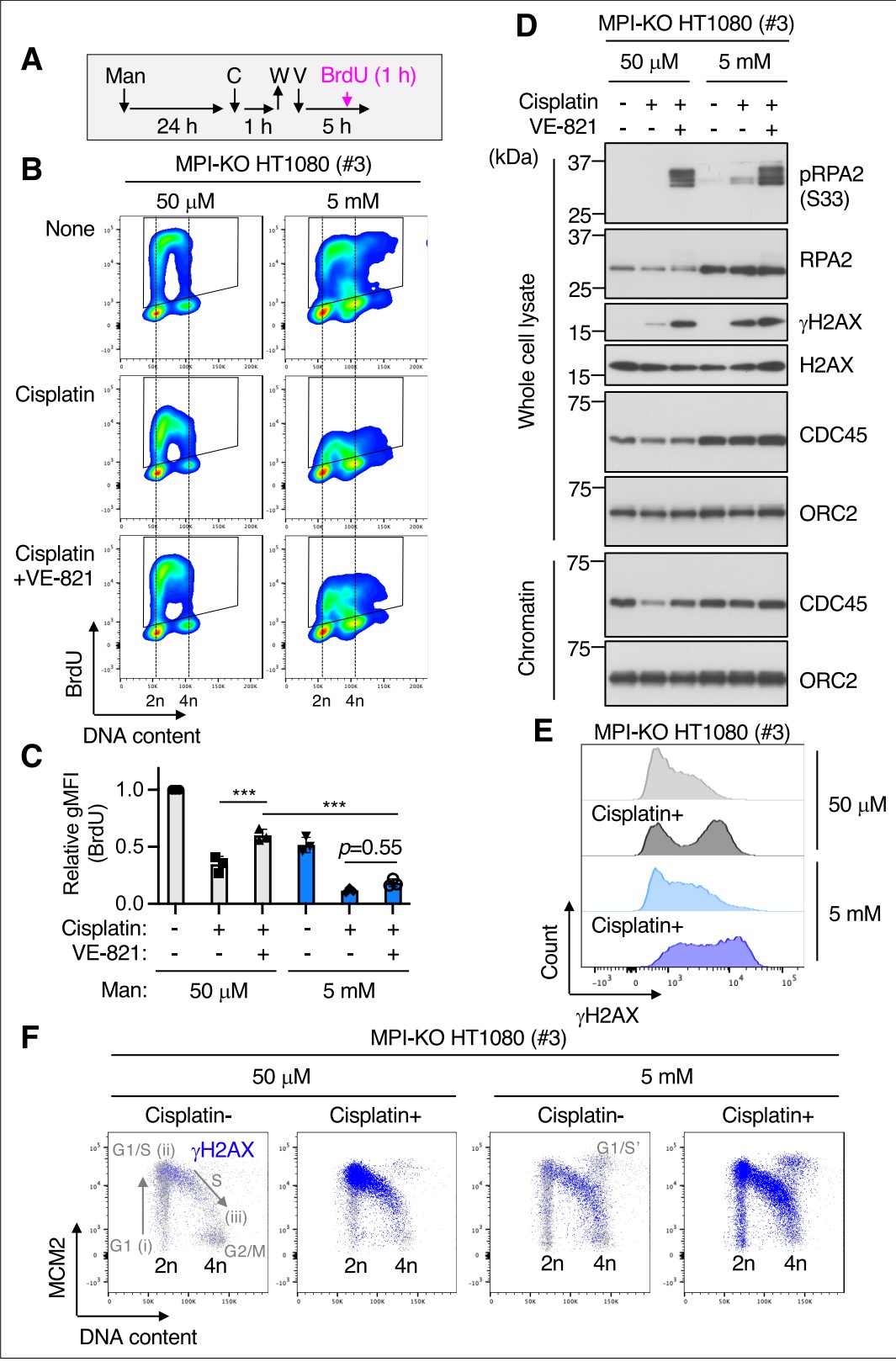

**Figure 4.** Mannose challenge disengages dormant origins from DNA synthesis during replication stress.
(**A**) Schematic representation of drug treatments. Man, 50 µM or 5 mM mannose; C, 100 µM cisplatin; W, wash; V, 1 µM VE-821; BrdU, 10 µM 5-bromo-2'-deoxyuridine. (**B**) Flow cytometry for BrdU and DNA content (Hoechst33342) in mannose phosphate isomerase knockout (MPI-KO) HT1080 (#3) cells treated as indicated in (**A**). The BrdU-

*Figure 4 continued on next page*

*Figure 4 continued*

positive cells were gated in black boxes for quantification of the geometric mean fluorescent intensity. (**C**) Relative geometric mean fluorescent intensity (gMFI) of BrdU in (**B**) (in the gated populations). The gMFI of BrdU in cells treated with 50 µM mannose in the absence of cisplatin (None) was set to 1.0. (**D**) Western blot analysis of whole-cell lysates and chromatin fractions of MPI-KO HT1080 (#3) cells treated as indicated in (**A**), except that BrdU labeling was omitted. (**E, F**) Chromatin flow cytometry for γH2AX, MCM2, and DNA content (Hoechst33342) in MPI-KO HT1080 (#3) cells treated as in (**A**), except that VE-821 treatment and BrdU labeling were omitted. The binding of MCM2 to chromatins (origin licensing) is detected in G1 phase (step i) and it peaks at G1/S boundary (step ii). As S phase progresses, MCM2 is dissociated from chromatins (step iii). $G_1/S'$ denotes the $G_1/S$ boundary of cell populations that underwent endoreplication. Data represent the mean ± SD; n = 3 independent experiments. ***p<0.001, one-way ANOVA with *post hoc* Tukey's test (**C**).

The online version of this article includes the following source data and figure supplement(s) for figure 4:

**Source data 1.** Original blot images depicting cropped regions for *Figure 4D*.

**Source data 2.** Original blot images depicting cropped regions for *Figure 4D*.

**Source data 3.** Original blot images depicting cropped regions for *Figure 4D*.

**Source data 4.** Original blot images depicting cropped regions for *Figure 4D*.

**Figure supplement 1.** Mannose challenge disengages dormant origins from DNA synthesis during replication stress in mannose phosphate isomerase knockout (MPI-KO) HeLa cells.

**Figure supplement 2.** Mannose challenge impairs loading/unloading dynamics of CDC45 and enhances cisplatin-induced genomic instability.

**Figure supplement 3.** The effects of mannose on cell proliferation and genomic instability in human cancer cells with varying levels of mannose phosphate isomerase (MPI).

**Figure supplement 3—source data 1.** Original blot images depicting cropped regions for *Figure 4—figure supplement 3A*.

**Figure supplement 3—source data 2.** Original blot images depicting cropped regions for *Figure 4—figure supplement 3C*.

**Figure supplement 3—source data 3.** Original blot images depicting cropped regions for *Figure 4—figure supplement 3D*.

**Figure supplement 3—source data 4.** Original blot images depicting cropped regions for *Figure 4—figure supplement 3E*.

**Figure supplement 3—source data 5.** Original blot images depicting cropped regions for *Figure 4—figure supplement 3F*.

**Figure supplement 4.** Identification of γH2AX-positive cells by chromatin flow cytometry at cell cycle levels.

was not severely impaired by mannose challenge (*Figure 4D* and *Figure 4—figure supplement 2A and B*). Under mannose-unchallenged conditions, cisplatin treatment induced the unloading of CDC45 from chromatin, which was restored by the forced activation of dormant origins (*Figure 4D* and *Figure 4—figure supplement 2A and B*). However, mannose challenge abrogated the CDC45 unloading/reloading dynamics under replication stress conditions (*Figure 4D* and *Figure 4—figure supplement 2A and B*).

The deficiency in the DNA synthesis from dormant origins and the CDC45 dynamics was associated with the increase in cisplatin-induced γH2AX (*Figure 4D and E* and *Figure 4—figure supplement 2C*), the phosphorylated form of the histone H2AX that marks DNA double-strand breaks (*Kuo and Yang, 2008*). Notably, mannose challenge did not enhance cisplatin-induced γH2AX in human cancer cell lines that express varying levels of MPI, although mannose challenge significantly suppressed cell proliferation in MPI[low] cancer cells (*Figure 4—figure supplement 3A–F*) as reported previously (*Gonzalez et al., 2018*). To identify which phase(s) of the cell cycle is associated with γH2AX positivity (γH2AX[+]), we performed triple-staining chromatin flow cytometry for MCM2, γH2AX, and DNA content (*Figure 4—figure supplement 4A*). Total single cells were used to plot total chromatin-bound MCM2 against DNA content, while γH2AX[+] populations were gated from single cells and used to plot the chromatin-bound MCM2 profiles against DNA content. These two plots were overlaid to visualize γH2AX[+] cells in each phase of the cell cycle. The endogenous levels of γH2AX were detected in S phase of both mannose-challenged and -unchallenged MPI-KO HT1080 and MPI-KO HeLa cells (*Figure 4F* and *Figure 4—figure supplement 4B*), and cisplatin strongly induced γH2AX throughout

S phase even in these mannose-challenged MPI-KO cancer cells (*Figure 4F* and *Figure 4—figure supplement 4B*). Together, these results indicate that mannose challenge disengages dormant origins from DNA synthesis during replication stress, thus exacerbating DNA damage in MPI-KO cancer cells.

## Mannose challenge causes bioenergetic imbalance and ATP insufficiency

Although our data showed that mannose challenge generated slow-cycling cells that failed to engage dormant origins in DNA synthesis during replication stress, the underlying mechanism that functionally links these two phenotypes was still unclear. Since cell cycle progression and DNA replication are metabolically demanding processes (*Zylstra and Heinemann, 2022*), we hypothesized that the metabolic checkpoints activated by honeybee syndrome (*DeRossi et al., 2006*; *Sols et al., 1960*) may play a key role in the anticancer activity of mannose. To test this hypothesis, we first examined the impact of mannose challenge on the cellular bioenergetics in MPI-KO HT1080 cells by monitoring the real-time changes in glycolytic flux in the form of extracellular acidification rate (ECAR) and the activity of oxidative phosphorylation (OXPHOS) in the form of oxygen consumption rate (OCR). Mannose challenge caused a steep drop of ECAR with a faint increase in OCR (*Figure 5A and B*), resulting in a marked reduction in ATP production rates (*Figure 5C*). The remaining ECAR further decreased after oligomycin A treatment in mannose-challenged cells, while the same treatment increased ECAR in mannose-unchallenged cells (*Figure 5A*), indicating that mannose challenge ablates glycolytic capacity, which is required to buffer the defects in OXPHOS. Consistent with these bioenergetic estimations, mannose challenge decreased the ATP pool in MPI-KO HT1080 cells, MPI-KO HeLa cells, and MPI-KO MEFs (*Figure 5D and E* and *Figure 5—figure supplement 1A and B*), and the remaining pool was almost completely depleted by co-treatment with IACS-010759, a preclinical small-molecule inhibitor of complex I of the mitochondrial respiratory chain (*Molina et al., 2018*; *Figure 5D and E* and *Figure 5—figure supplement 1A and B*), which in turn increased necrosis (*Figure 5F*). These results indicate that mannose-challenged cells highly depend on OXPHOS for cellular bioenergetics.

## Mannose challenge generates a distinct metabolic landscape

To provide deeper insights into the metabolic landscape generated by honeybee syndrome, we compared the metabolome between mannose-challenged and -unchallenged MPI-KO HT1080 cells (*Figure 5—figure supplement 2A*, *Source data 3*). One-day mannose challenge greatly increased the pool of Fruc-6-P, while that of the three-carbon metabolites in the lower glycolysis chain, including lactate, were substantially decreased (*Figure 5G*), supporting our earlier findings that mannose challenge decreased ECAR to the basal level. Despite this glycolytic alteration, the steady state levels of metabolites in oxidative and non-oxidative arms of the pentose phosphate pathway (PPP) remained relatively unchanged in the mannose-challenged cells, except for the large accumulation of 6-phosphogluconoate and the slight decrease in sedoheptulose-7-phosphate (*Figure 5H*). In contrast, mannose challenge severely decreased tricarboxylic acid (TCA) cycle intermediates (isocitrate, 2-oxoglutarate, succinate, fumarate, and malate) that are used to generate NADH and $FADH_2$ as electron donors for the mitochondrial respiratory chain (*Figure 5—figure supplement 2B*). Strikingly, the amounts of ribonucleoside diphosphates and triphosphates, but not ribonucleoside monophosphates, were proportionally and moderately decreased (*Figure 5I–K*), whereas deoxyribonucleoside triphosphates (dNTPs) were substantially decreased in the mannose-challenged cells (*Figure 5L*). Collectively, these results indicate that mannose challenge generates a distinct metabolic landscape in MPI-KO HT1080 cells, which ultimately lead to the depletion of dNTP pools.

## Mannose challenge severely reduces the capacity for biosynthesizing nucleotides

dNTPs are the essential donor substrates for DNA synthesis, raising the possibility that the dNTP loss caused by mannose challenge may be a major mechanism linking the generation of slow-cycling cells and the failure to engage dormant origins in DNA synthesis during replication stress. To further elucidate the impacts of mannose challenge on the biosynthesis of dNTPs, we performed $[^{13}C_6]$-glucose tracer experiments that, unlike steady-state metabolomics, allow to estimate the activity of glucose-related metabolic pathways by analyzing the fractional enrichment of the metabolites in a dynamic labeling phase (*Lorkiewicz et al., 2019*). The MPI-KO HT1080 cells were preconditioned under

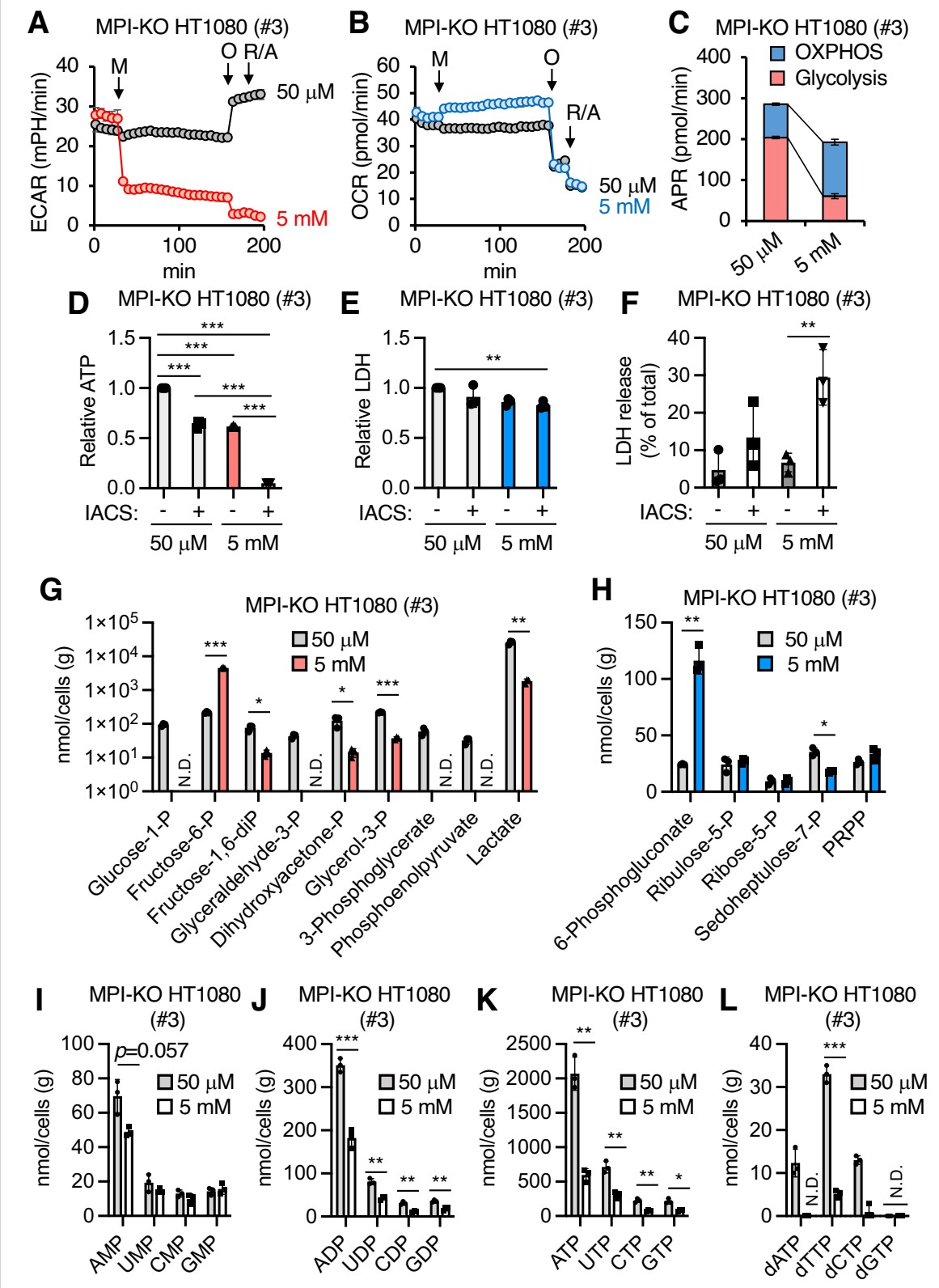

**Figure 5.** Mannose challenge impairs bioenergetic balance and generates a distinct metabolic landscape. (**A, B**) Extracellular acidification rates (ECAR, **A**) and oxygen consumption rates (OCR, **B**) in mannose phosphate isomerase knockout (MPI-KO) HT1080 (#3) cells. The cells were first cultured in 50 μM mannose, and then the medium alone or mannose (5 mM final, M) was added to the cells. They were incubated for 120 min and further treated with 1.5 μM oligomycin A (O) for 18 min, followed by 0.5 μM each of rotenone and antimycin A (R/A) for 18 min. (**C**) ATP production rates (APR) estimated

*Figure 5 continued on next page*

*Figure 5 continued*

from the data in (**A**, **B**). See also the 'Materials and methods.' (**D**) Relative ATP levels in MPI-KO HT1080 (#3) cells treated for 6 hr with or without 30 nM IACS-010759 (IACS) in the presence of 50 µM or 5 mM mannose. (**E**) Relative lactate dehydrogenase (LDH) levels in MPI-KO HT1080 (#3) cells treated as in (**D**). (**F**) LDH release from MPI-KO HT1080 (#3) cells treated for 24 hr with or without 30 nM IACS-010759 (IACS) in the presence of 50 µM or 5 mM mannose. (**G–L**) Metabolomic profiling of glycolysis (**G**), the pentose phosphate pathway (**H**), ribonucleotides (**I–K**), and deoxyribonucleoside triphosphates (**L**) in MPI-KO HT1080 (#3) cells cultured for 24 hr in the presence of 50 µM or 5 mM mannose. P, phosphate; PRPP, 5-phosphoribosyl-1-pyrophosphate. Data represent the mean ± SD; n = 3 independent experiments. *p<0.05, **p<0.01, and ***p<0.001, one-way ANOVA with *post hoc* Tukey's test (**D–F**) or Welch's *t*-test (**G–L**).

The online version of this article includes the following figure supplement(s) for figure 5:

**Figure supplement 1.** Mannose-challenged mannose phosphate isomerase knockout (MPI-KO) cells highly depend on oxidative phosphorylation (OXPHOS) for ATP production.

**Figure supplement 2.** Metabolic changes in mannose-challenged and -unchallenged mannose phosphate isomerase knockout (MPI-KO) HT1080 cells.

mannose-challenged or -unchallenged conditions before [$^{13}C_6$]-glucose labeling. The $^{13}$C-labeling of most metabolites detected in glycolysis, the PPP, and the TCA cycle (*Figure 6—figure supplement 1A*) already reached isotopic steady states within 30 min in both mannose-challenged and -unchallenged cells (*Figure 6—figure supplement 1B*), most likely due to their high metabolic activity (*Jang et al., 2018*; *Lorkiewicz et al., 2019*). In contrast to these central metabolic pathways, metabolites in purine and pyrimidine metabolism showed a dynamic or sub-dynamic labeling in mannose-challenged cells. Phosphoribosyl pyrophosphate (PRPP) is an essential ribose donor substrate for the biosynthesis of nucleotides in both purine and pyrimidine metabolism (*Figure 6A and B* and *Figure 6—figure supplement 2A*). Fractional enrichment of $^{13}$C-PRPP (M+5) was saturated at 30 min in unchallenged cells, while it was still in a sub-dynamic labeling phase in mannose-challenged cells (*Figure 6C*). Despite this slower fractional enrichment, the pool size of $^{13}$C-PRPP (M+5) in mannose-challenged cells was similar to that in unchallenged cells (*Figure 6D*), implying that mannose challenge reduced the utilization of PRPP for nucleotide biosynthesis. Consistent with this assumption, we found a marked reduction in both the fractional enrichment and the pool size of $^{13}$C-purine metabolic intermediates (M+5) in mannose-challenged cells, which included inosine-5'-monophosphate (IMP; *Figure 6E and F*), adenosine-5'-monophosphate (AMP; *Figure 6G and H*), and guanosine-5'-monophosphate (GMP; *Figure 6I and J*).

PRPP is utilized for both de novo and salvage synthesis of purine nucleotides (*Figure 6A*), and therefore the M+5 fraction of $^{13}$C-IMP, $^{13}$C-AMP, and $^{13}$C-GMP can be accounted for the sum of their de novo and salvage pools. In contrast, purine nucleotides with the labeled fractions greater than M+5 can originate from de novo synthesis (*Figure 6A*). We found a progressive increase in the M+6 fraction of $^{13}$C-IMP, $^{13}$C-AMP, and $^{13}$C-GMP in unchallenged cells (*Figure 6E, G and I*), suggesting that $^{13}$C-10-formyl-tetrahydrofolate (CHO-THF; M+1), which is produced de novo via the glycolysis-serine biosynthesis-folate cycle (GSF) axis (*Figure 6A*; *Yang and Vousden, 2016*), contributed to the de novo synthesis of purine nucleotides. Although we could not directly detect $^{13}$C-labeling in serine and CHO-THF in our metabolomic analysis, the fractional enrichment and the pool size of $^{13}$C-glycine (M+2), which is a signature metabolite produced in coupled with 5,10-methylenetetrahydrofolate via the GSF axis, progressively increased in unchallenged cells (*Figure 6K and L*). However, the fractional enrichment and the pool size of $^{13}$C-glycine (M+2) largely decreased in mannose-challenged cells (*Figure 6K and L*), suggesting that the GSF axis is compromised in these cells. These results may partly explain why the de novo synthesis of purine nucleotides is limited in honeybee syndrome.

In the early stage of pyrimidine metabolism, aspartate (Asp) is transferred to carbamoyl phosphate, giving rise to *N*-carbamoyl aspartate (carbamoyl Asp) (*Figure 6—figure supplement 2A*). We found a progressive increase in the M+2, M+3, and M+4 fractions of both $^{13}$C-Asp (*Figure 6—figure supplement 2B and C*) and $^{13}$C-carbamoyl Asp (*Figure 6—figure supplement 2D and E*) at similar labeling rates in unchallenged cells. The $^{13}$C-Asp (M+2, M+3, and M+4) could originate from [$^{13}C_6$]-glucose-derived oxaloacetate that is formed in the first, second, and third rounds of the TCA cycle (*Figure 6—figure supplement 2F*), as indicated by the presence of the M+2, M+3, and M+4 fractions of $^{13}$C-malate (*Figure 6—figure supplement 2G and H*). However, mannose challenge severely decreased the fractional enrichment of both $^{13}$C-Asp (M+2, M+3, and M+4) and $^{13}$C-carbamoyl Asp (M+2, M+3, and M+4) (*Figure 6—figure supplement 2B and D*). In contrast, the pool size of $^{13}$C-Asp (M+2) remained relatively unchanged between mannose-challenged and -unchallenged cells

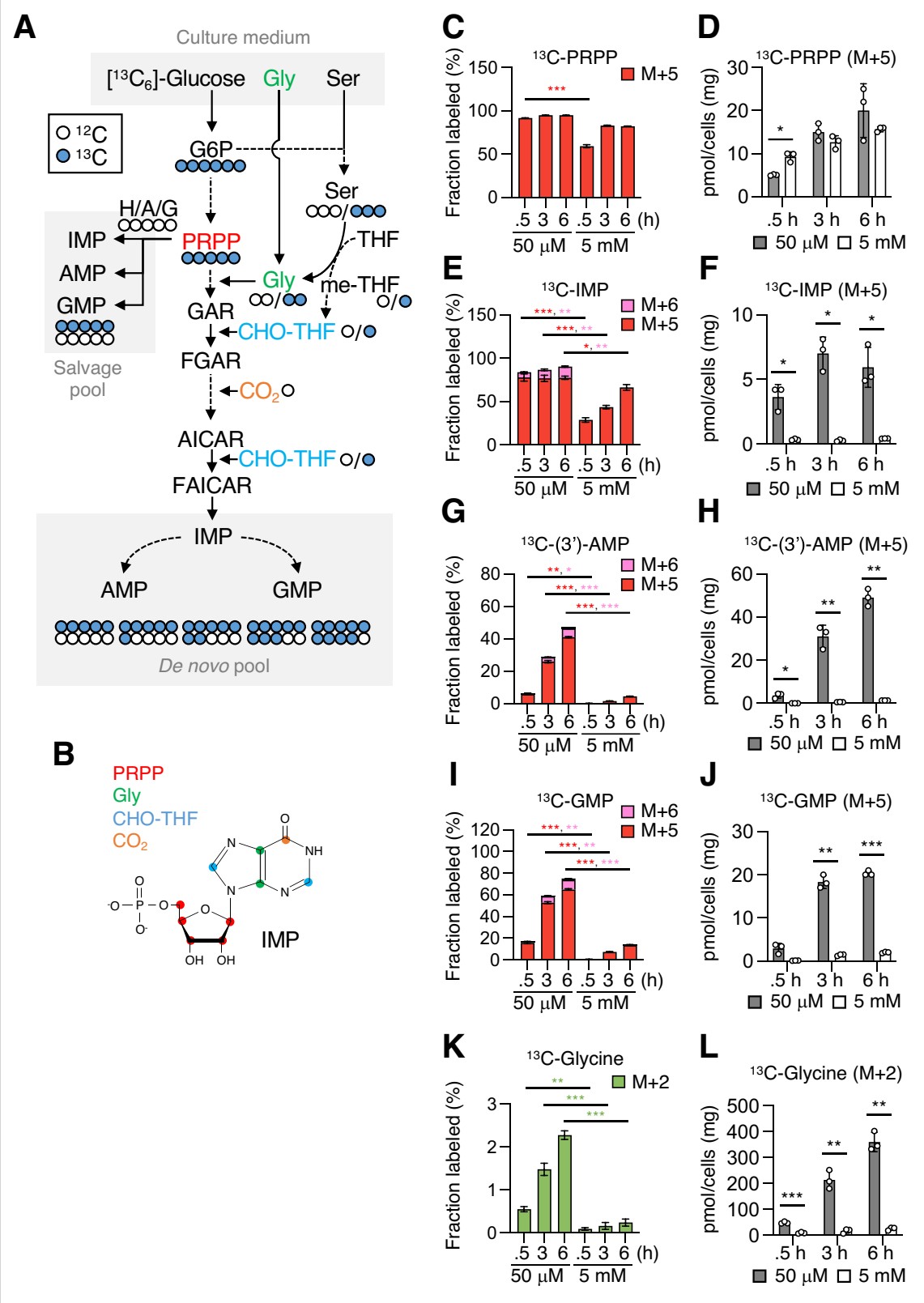

**Figure 6.** Mannose challenge severely impairs purine metabolism. (**A**) A schematic representation of [$^{13}C_6$]-glucose tracing in purine metabolism. Blue and white circles denote $^{13}C$ and $^{12}C$, respectively. Phosphoribosyl pyrophosphate (PRPP) is produced from glucose 6-phosphate (G6P) through glycolysis and the pentose phosphate pathway, while serine (Ser), glycine (Gly), 5,10-methylenetetrahydrofolate (me-THF), and 10-fomyltetrahydrofolate (CHO-THF) are produced through the glycolysis-serine biosynthesis-folate cycle axis. In salvage pathway of purine metabolism, PRPP is conjugated

*Figure 6 continued on next page*

*Figure 6 continued*

with hypoxanthine (X), adenine (A), or guanine (G) to form inosine 5'-monophosphate (IMP), adenosine 5'-monophosphate (AMP), or guanosine 5'-monophosphate (GMP), respectively. In the de novo pathway, IMP is formed from PRPP via multiple steps and used to produce AMP and GMP. Possible labeling patterns with $^{12}$C and $^{13}$C were indicated. GAR, 5'-phosphoribosylglycinamide; FGAR, 5'-phosphoribosyl-*N*-formylglycinamide; AICAR, 5'-phosphoribosyl-5-amino-4-imidazolecarboxamide; FAICAR, 5'-phosphoribosyl-5-formamido-4-imidazolecarboxamide. (**B**) A molecular structure of IMP. Red, green, blue and orange circles indicate carbon atoms originating from PRPP, Gly, CHO-THF and $CO_2$, respectively. (**C−L**) Fractional enrichment (fraction labeled; **C, E, G, I, K**) and pool size (**D, F, H, J, L**) of the indicated metabolites. The mannose phosphate isomerase knockout (MPI-KO) HT1080 (#3) cells were preconditioned in mannose-challenged (5 mM) or -unchallenged (50 µM) culture medium for 24 hr and then labeled with [$^{13}C_6$]-glucose for the indicated periods before harvest. The amounts of AMP were estimated as a mixture with 3'-AMP [(3')-AMP] due to their insufficient separation. Data represent the mean ± SD; n = 3 independent experiments. *p<0.05, **p<0.01, ***p<0.001, Welch's *t*-test.

The online version of this article includes the following figure supplement(s) for figure 6:

**Figure supplement 1.** [$^{13}C_6$]-Glucose-tracing analysis for metabolites in glycolysis, the pentose phosphate pathway (PPP), and the tricarboxylic acid (TCA) cycle.

**Figure supplement 2.** [$^{13}C_6$]-Glucose-tracing analysis for metabolites in pyrimidine metabolism.

**Figure supplement 3.** [$^{13}C_6$]-Glucose-tracing analysis for deoxyribonucleoside triphosphates (dNTPs).

**Figure supplement 4.** Supplementation of mannose-challenged mannose phosphate isomerase knockout (MPI-KO) HT1080 cells with hypoxanthine and thymidine or with deoxyribonucleoside mixture does not improve cell proliferation.

(*Figure 6—figure supplement 2C*), while the pool size of $^{13}$C-carbamoyl Asp (M+2) greatly decreased in mannose-challenged cells (*Figure 6—figure supplement 2E*), indicating that mannose challenge reduced the utilization of Asp in pyrimidine metabolism. In the immediate downstream metabolites of carbamoyl Asp, we could detect significant amounts of uridine-5'-monophosphate (UMP) and found that a large majority of $^{13}$C-UMP formed in unchallenged cells was consisted of the M+5 fraction (*Figure 6—figure supplement 2I and J*). This fraction was most likely to originate from unlabeled carbamoyl Asp (*Figure 6—figure supplement 2K*) and $^{13}$C-PRPP (M+5). Moreover, we identified the M+6, M+7, and M+8 fractions of $^{13}$C-UMP in unchallenged cells (*Figure 6—figure supplement 2I*), indicating that both $^{13}$C-carbamoyl Asp (M+2, M+3, and M+4) and $^{13}$C-PRPP (M+5) contributed to forming $^{13}$C-orotidine-5'-monophosphate (M+7, M+8, and M+9), which is decarboxylated to give $^{13}$C-UMP (M+6, M+7, and M+8). However, mannose-challenged cells showed a substantial reduction of $^{13}$C-UMP in both the fractional enrichment (M+5, M+6, M+7, and M+8) and the pool size (M+5) (*Figure 6—figure supplement 2I and J*), clearly indicating that mannose challenge impaired the biosynthesis of pyrimidine nucleotides. As expected, dNTPs showed little $^{13}$C enrichment (dATP and dGTP, *Figure 6—figure supplement 3A–E*) and a very low $^{13}$C enrichment (dTTP and dCTP, *Figure 6—figure supplement 3F–J*) in mannose-challenged cells compared with those in unchallenged cells. Taking all these findings together, mannose challenge impairs both purine and pyrimidine metabolism at the early stage, thereby potentially limiting the de novo synthesis of dNTPs.

## Pharmacological inhibition of de novo dNTP biosynthesis retards cell cycle progression, increases chemosensitivity, and inhibits DNA synthesis from dormant origins

To examine whether the loss of dNTPs plays a key role in the anticancer activity of mannose, we inhibited de novo synthesis of dNTPs with hydroxyurea (HU), a highly potent inhibitor of ribonucleotide reductase (RNR) (*Elford, 1968*). We found that the combination of HU and cisplatin more severely reduced cell viability than cisplatin treatment alone, independently of the presence or absence of the *MPI* gene (*Figure 7A*). This chemosensitizing effect of HU was associated with the strong induction of γH2AX (*Figure 7B*). HU is widely used as an agent to arrest cells in S phase by depleting dNTP (*Davis et al., 2001*). Profiling of the chromatin-bound MCM2 showed that HU treatment at a moderate concentration (0.25 mM) resulted in the arrest of cells in the early to late S phase, while a higher concentration of HU (1 mM) almost completely arrested the cells at the $G_1$/S boundary (*Figure 7C*), partly recapitulating the inhibitory effects of mannose challenge on cell cycle progression. As with the case of mannose challenge, the forced activation of dormant origins by ATRi failed to increase BrdU uptake in the presence of HU (*Figure 7D–G*), indicating that the DNA synthesis from dormant origins is highly sensitive to the pool size of dNTPs. Unlike mannose challenge, however, HU treatment did not severely impair the CDC45 unloading/reloading dynamics during replication stress (*Figure 7G*),

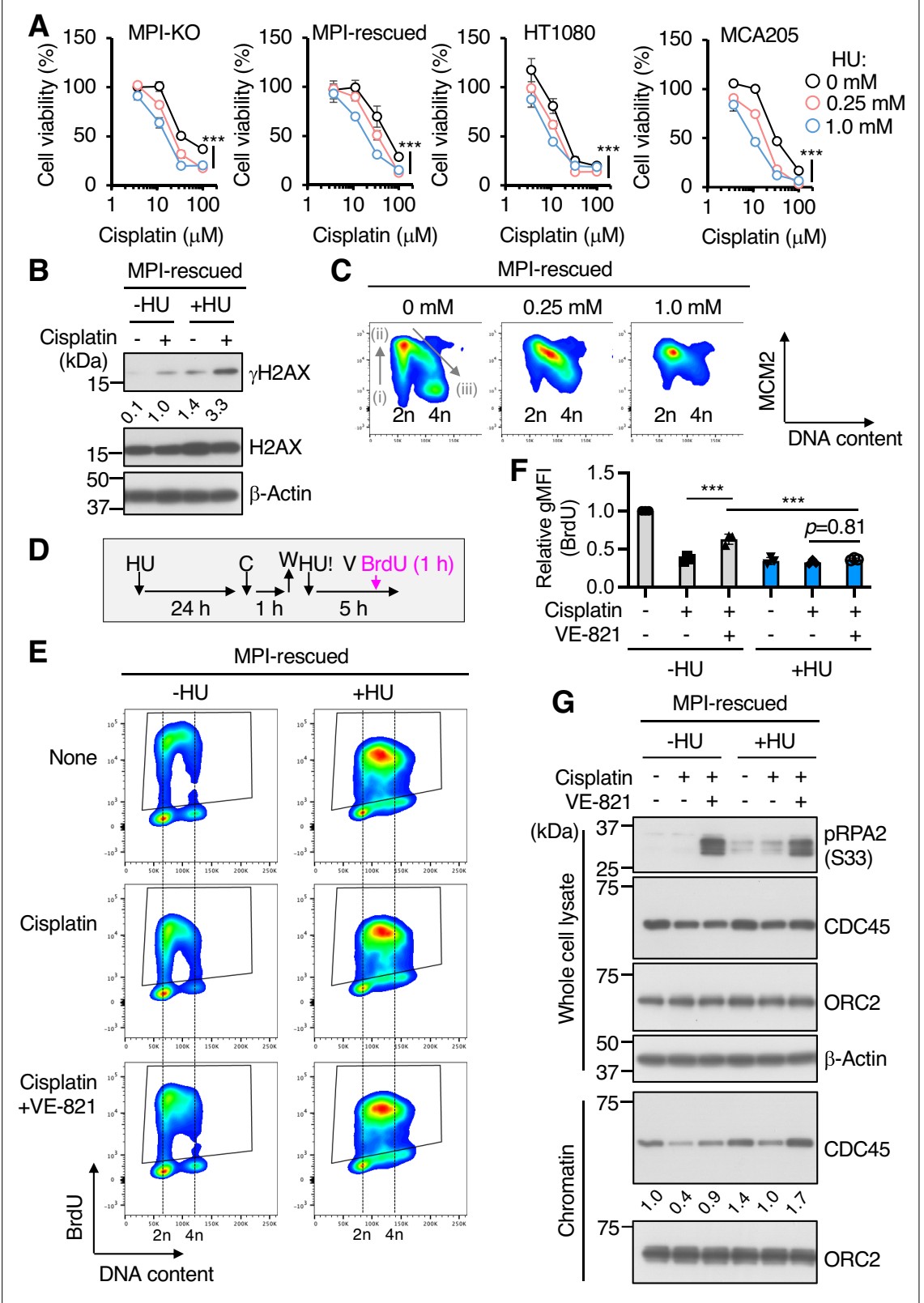

**Figure 7.** Pharmacological inhibition of de novo deoxyribonucleoside triphosphate (dNTP) biosynthesis retards cell cycle progression, increases chemosensitivity, and inhibits DNA synthesis from dormant origins. (**A**) Cell viability assay in mannose phosphate isomerase knockout (MPI-KO) HT1080 (#3), the *MPI*-rescued cells (*MPI*-rescued), the parental HT1080, and mouse fibrosarcoma MCA205 cells that were preconditioned with hydroxyurea (HU) for 24 hr, followed by incubation with cisplatin for an additional 24 hr in the presence of the same concentrations of HU used for preconditioning.

*Figure 7 continued on next page*

*Figure 7 continued*

(**B**) Western blot analysis of whole-cell lysates from the *MPI*-rescued MPI-KO HT1080 (#3) cells preconditioned with 0.25 mM HU for 24 hr and pulsed with 100 µM cisplatin for 1 hr, followed by a 5 hr chase. Numbers indicate the relative amounts of γH2AX (the average of two independent experiments each from the *MPI*-rescued MPI-KO HT1080 cells and the parental HT1080 cells). (**C**) Chromatin flow cytometry for MCM2 and DNA content (Hoechst33342) in the *MPI*-rescued MPI-KO HT1080 (#3) cells treated with or without HU for 24 hr at the indicated concentrations. The binding of MCM2 to chromatin is detected in G1 phase (step i) and it peaks at G1/S boundary (step ii). As S phase progresses, MCM2 is dissociated from chromatin (step iii). (**D**) Schematic representation of drug treatment. HU, 0.25 mM hydroxyurea; C, 100 µM cisplatin; W, wash; V, 1 µM VE-821; 10 µM BrdU, 5-bromo-2'-deoxyuridine. (**E**) Flow cytometry for BrdU and DNA content (Hoechst33342) in the *MPI*-rescued MPI-KO HT1080 (#3) cells that were treated as indicated in (**D**). The BrdU-positive cells were gated in black boxes for quantification of the geometric mean fluorescent intensity. (**F**) Relative geometric mean fluorescent intensity (gMFI) of BrdU in (**E**) (in the gated populations). The gMFI of BrdU in cells treated without HU (None) was set to 1.0. (**G**) Western blot analysis of whole-cell lysates and chromatin fractions of the *MPI*-rescued MPI-KO HT1080 (#3) cells treated as indicated in (**D**), except that BrdU labeling was omitted. Numbers indicate the relative amounts of chromatin-bound CDC45 (the average of three independent experiments). Data represent the mean ± SD; n = 3 independent experiments. *p<0.05, **p<0.01, and ***p<0.001, two-way ANOVA with *post hoc* Bonferroni's test (**A**) or one-way ANOVA (**F**) with *post hoc* Tukey's test.

The online version of this article includes the following source data for figure 7:

**Source data 1.** Original blot images depicting cropped regions for *Figure 7B*.

**Source data 2.** Original blot images depicting cropped regions for *Figure 7G*.

**Source data 3.** Original blot images depicting cropped regions for *Figure 7G*.

**Source data 4.** Original blot images depicting cropped regions for *Figure 7G*.

indicating that the insufficient dNTP pool is a major cause of the disengagement of dormant origins from DNA synthesis.

## Discussion

MPI is the sole enzyme that catalyzes the interconversion between Fruc-6-P and Man-6-P in mammals. The conversion of Fruc-6-P to Man-6-P mediated by MPI is central to the synthesis of GDP-mannose from abundant glucose for normal glycosylation. In contrast, the conversion of Man-6-P to Fruc-6-P mediated by the same enzyme is critical to directing the excess Man-6-P to glycolysis, while the

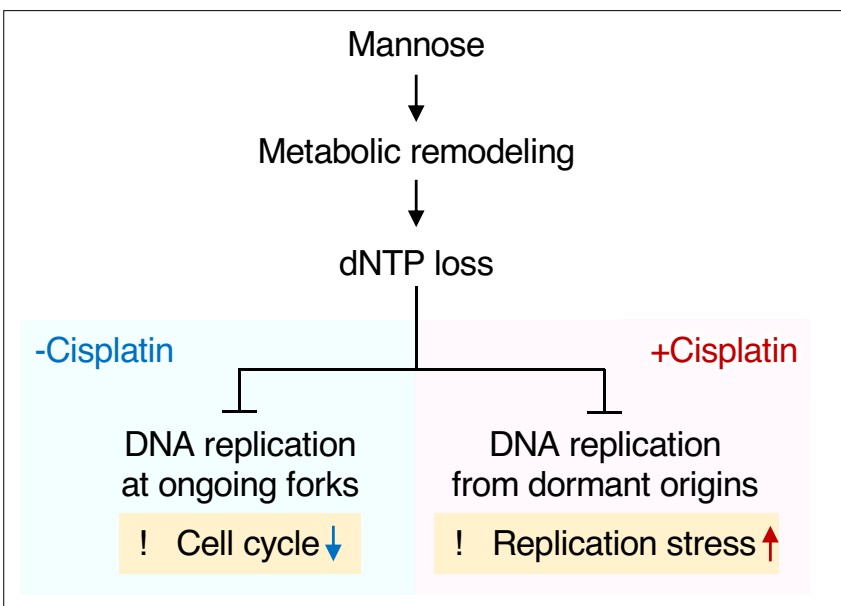

**Figure 8.** Proposed model for the mechanism underlying anticancer activity of mannose in mannose phosphate isomerase knockout (MPI-KO) human cancer cells. A large influx of mannose into MPI-KO human cancer cells induces metabolic remodeling, leading to the loss of deoxyribonucleoside triphosphates (dNTPs). This metabolic deficiency impairs DNA replication at ongoing replication forks and slows cell cycle progression in the absence of cisplatin (-Cisplatin). On the other hand, mannose-caused dNTP loss disengages dormant origins from DNA replication in the presence of cisplatin, thereby exacerbating replication stress (+Cisplatin).

cellular function of this seemingly wasteful metabolic pathway has long remained unknown. In this study, we employed MPI-KO human cancer cells to explore the key mechanism behind the anticancer activity of mannose and demonstrated that the large influx of mannose exceeding the capacity to metabolize it, that is, the onset of honeybee syndrome, induced dramatic metabolic remodeling that led to ATP insufficiency and dNTP loss (*Figure 8*). These cells were cycling extremely slowly, and upon encountering replication stress, they were unable to rescue stalled forks via dormant origins, thus exacerbating replication stress (*Figure 8*). These findings shed light on the conversion of Man-6-P to Fruc-6-P mediated by MPI as a genome guard through the maintenance of metabolic integrity, which could be threatened by dietary mannose intake.

The induction of ATP insufficiency by honeybee syndrome coincided with that of abnormal cell cycle progression, indicating the critical role of normal mannose catabolism in balancing bioenergetic states and cell cycle progression. This finding is supported by the accumulating evidence indicating that excess mannose inhibits glycolysis and retards cell proliferation in MPI-KO MEFs (*DeRossi et al., 2006*) and MPI$^{low}$ cancer cells (*Gonzalez et al., 2018*), and that the temporal regulation of glycolysis and OXPHOS drives the G$_1$/S transition and chromosome segregation (*Icard et al., 2019*; *Salazar-Roa and Malumbres, 2017*). Moreover, the fact that HU treatment inhibits the progression of replication forks suggests that dNTP loss in honeybee syndrome impairs S phase progression. Taking these findings together, the massive changes in crosstalk between metabolic networks and cell cycle machinery in honeybee syndrome potentially contribute to cell cycle dysregulation, which may further complicate metabolic states (*Fajas, 2013*).

An examination of human cancer cell lines that express varying levels of MPI clearly indicated that the antiproliferative effect of mannose is largely dependent on the expression levels of MPI, while mannose-driven genomic instability is a unique phenotype observed in the absence of MPI. For this reason, the potential clinical use of mannose to induce genomic instability in cancer cells is currently unlikely. Here, we provide experimental evidence showing that pharmacological inhibition of de novo dNTP synthesis, which is critical for the recovery from replication stress (*Håkansson et al., 2006*; *Tanaka et al., 2000*), partly mimics the anticancer activity of mannose and chemosensitizes cells to cisplatin independently of the MPI expression and mannose dosing. Our finding that DNA synthesis from dormant origins during replication stress is highly sensitive to the dNTP pool size is in good agreement with the therapeutic advantages of RNR inhibition in enhancing the efficacy of radiochemotherapy (*Kunos and Ivy, 2018*). These findings strongly suggest dNTP loss as an important mechanism underlying the chemosensitizing effect of mannose in MPI-KO cancer cells.

What causes the loss of dNTPs in honeybee syndrome? The pool size of dNTPs is tightly regulated in order to meet and not to exceed cellular demands by cell cycle-dependent and DNA damage-dependent expression of enzymes involved in the biosynthesis and degradation of dNTPs (*Franzolin et al., 2013*), as well as their allosteric regulation (*Aye et al., 2015*; *Ji et al., 2013*). This tight regulation of dNTP supply is vital to prevent cells from suffering the dNTP pool imbalance, which causes genomic instability (*Kumar et al., 2010*; *Pajalunga et al., 2017*). Our proteomic analysis revealed that mannose challenge rapidly increases the expression of RRM2B (*Source data 1*), which is a rate-limiting and TP53-inducible subunit of RNR (*Håkansson et al., 2006*; *Tanaka et al., 2000*), indicating that the equilibrium of dNTP metabolism shifted toward biosynthesis at the protein level. However, mannose challenge caused substantial reductions in the pool of ATP, which is a key allosteric regulator for the catalysis of RNR to occur (*Brignole et al., 2018*). Moreover, a recent in-depth metabolomic study revealed that a high concentration of mannose impairs glucose metabolism in MPI$^{low}$ cancer cells (*Gonzalez et al., 2018*), while its impact on the downstream nucleotide metabolism was not clear. Our [$^{13}$C$_6$]-glucose tracer experiments clearly indicated that mannose challenge severely impairs the early stages of de novo synthesis of purine and pyrimidine nucleotides. Fueling the salvage pathway of purine and pyrimidine metabolism with hypoxanthine and thymidine (*Figure 6—figure supplement 4A*) or with deoxyribonucleoside mixture (*Figure 6—figure supplement 4B*) was not sufficient to improve the defects in proliferation of mannose-challenged MPI-KO HT1080 cells. Thus, the inadequate de novo synthesis of nucleotides in honeybee syndrome may limit dNTP pools in this cell model. Notably, although dNTPs are essential for DNA repair, the chemosensitizing effect of mannose was rather modest when comparing it with that of deficiency in genes for DNA repair processes (e.g., *BRCA2*, *FANCA*, and *FANCD2*) (*Bruno et al., 2017*; *Sakai et al., 2008*). This apparent discrepancy most likely arises from the fact that the suppression of de novo nucleotide

biosynthesis in honeybee syndrome is incomplete. Taken together, our findings indicate that the insufficient metabolic activity of purine and pyrimidine metabolism in honeybee syndrome causes the loss of dNTPs.

Our proteomic analysis revealed the functional pathways affected by honeybee syndrome. The downregulation of MCM2-7 proteins in mannose-challenged cells supports our hypothesis that honeybee syndrome impairs dormant origins and causes genomic instability (*Ibarra et al., 2008*; *Kawabata et al., 2011*; *Shima et al., 2007*). Mannose challenge also decreased the expression of several large ribosomal subunit proteins. Ribosome biogenesis stress has been identified as a major mechanism by which oxaliplatin kills cancer cells in a DNA damage response-independent manner (*Bruno et al., 2017*). Moreover, mannose challenge upregulated proteins involved in ferroptosis and necroptosis. These two programmed cell death mechanisms are considered as an effective approach to overcome chemoresistance (*Zhang et al., 2022*) or to eradicate apoptosis-resistant cancer cells (*Su et al., 2016*). In addition to metabolic impacts, honeybee syndrome may also enhance cancer cell vulnerability at the proteomic level.

In conclusion, we demonstrate that honeybee syndrome triggers dNTP loss and genomic instability in MPI-KO human cancer cells. Ensuring appropriate amounts of dNTPs is essential for normal DNA replication, as well as for the recovery from replication stress, thus providing a plausible mechanism for how mannose sensitizes poorly proliferating cancer cells to chemotherapy. Elucidation of the precise molecular mechanism underlying dNTP loss in honeybee syndrome will be necessary to unambiguously identify the direct cause(s) of the chemosensitizing effects of mannose. Our findings also pointed to there being significant phenotypic diversity in honeybee syndrome. More extensive biochemical characterization of the metabolomic and proteomic alterations in this syndrome will be needed to further dissect the impact of mannose metabolic alterations in cancer cell homeostasis, which may provide an opportunity for mannose-based drug discovery and development for cancer therapy.

## Materials and methods
### Experimental models
The human fibrosarcoma cell line HT1080 used in this study was our laboratory stock and validated by short tandem repeat profiling (Promega). The human cervix adenocarcinoma cell line HeLa and human pancreatic ductal cell carcinoma cell line KP-4 were obtained from RIKEN BioResource Research Center. The human non-small cell lung carcinoma cell line A549 and human ovarian adenocarcinoma cell line SK-OV-3 were obtained from American Type Culture Collection. The mouse fibrosarcoma cell line MCA205 was a kind gift from Dr. S. A. Rosenberg (National Cancer Institute, Bethesda, MD). These cell lines were cultured in Dulbecco's modified Eagle's medium (DMEM; 045-30285, FUJIFILM Wako) supplemented with 4 mM glutamine (073-05391, FUJIFILM Wako) and 10% fetal bovine serum (FBS) (complete DMEM) at 37°C in a 5% $CO_2$ atmosphere. Immortalized wild-type MEFs and MPI-KO MEFs were prepared in a previous study (*DeRossi et al., 2006*). MPI-KO HT1080 cells and MPI-KO HeLa cells were established in this study (see below). All MPI-KO cell lines were maintained in complete DMEM supplemented with mannose (Man; 130–00872, FUJIFILM Wako) at a concentration of 20 µM at 37°C in a 5% (for MPI-KO HT1080 cells and MPI-KO HeLa cells) and 10% (for MEFs) $CO_2$ atmosphere. Trypan blue dye (0.4%, w/v, FUJIFILM Wako) was mixed with equal volumes of cell suspension for cell counting. All cell lines were routinely tested for Mycoplasma and confirmed to have no contamination using TaKaRa PCR Mycoplasma Detection Set (6601, Takara).

### Cell treatments
Cells were treated with 0–100 µM cisplatin (D3371, Tokyo Chemical Industry), 0–10 µM doxorubicin (040-21521, FUJIFILM Wako), 30 nM IACS-010759 (S8731, Selleckchem), 1 µM VE-821 (SML1415, Sigma-Aldrich), 1× HT supplement (11067030, Thermo Fisher Scientific), the deoxyribonucleoside mix (deoxyadenosine, thymidine, deoxyguanosine, and deoxycytidine; 10 µM each), or 0–1 mM HU (085-06653, FUJIFILM Wako) for the indicated time. As a negative control, cells were treated with vehicle alone.

## Gene editing

*MPI* gene was knocked out by CRISPR–Cas9 gene editing using the Edit-R system (Dharmacon), in accordance with the manufacturer's instructions. Briefly, HT1080 cells or HeLa cells were seeded and cultured for 24 hr prior to transfection. Synthetic CRISPR RNAs (CM-011729-01 and CM-011729-03, Dharmacon; 10 µM, 2 µL each) targeting the *MPI* gene were mixed with 4 µL of 10 µM trans-activating CRISPR RNA (tracrRNA; U-002005-05, Dharmacon) and transfected with 1 µg of Edit-R SMART Cas9_ mCMV_(PuroR) expression plasmid (U-005200-120, Dharmacon) using 5 µL of Lipofectamine 2000 (11668027, Thermo Fisher Scientific) in 400 µL of Opti-MEM (31985062, Thermo Fisher Scientific). Transfectants were selected in complete medium supplemented with 50 µM Man and 1 µg/mL puromycin, and then cloned by limiting dilution. For HT1080 cells, MPI-KO clones were screened by polymerase chain reaction (PCR) using genomic DNA as a template and two primer sets (*Supplementary file 1*) for amplifying the wild-type allele and for the KO allele of the *MPI* locus. The PCR products for the KO allele were subjected to Sanger sequencing. For HeLa cells, MPI-KO clones were screened by mannose auxotrophy and sensitivity, as well as western blotting using anti-MPI antibody (see below).

## Preparation of cDNA

Total RNA was prepared using TRIzol Reagent (15596026, Thermo Fisher Scientific) or RNeasy Mini Kit (74106, QIAGEN), in accordance with the manufacturer's instructions. cDNA was prepared by reverse transcription using total RNA (4 µg), oligo dT primers, and SuperScript IV Reverse Transcriptase (18090010, Thermo Fisher Scientific), in accordance with the manufacturer's instructions.

## Preparation of plasmids

The plasmids used in this study are listed in *Supplementary file 2*. The coding sequence of the human *MPI* gene was amplified by PCR using Phusion High-Fidelity DNA polymerase (M0530, New England BioLabs), a primer set (*Supplementary file 1*), and cDNA from HuH-7 cells as a template. The PCR products were purified from agarose gels using NucleoSpin Gel and PCR Clean-Up Kit (740609, Clontech) and subcloned into the pENTR/D-TOPO vector (K240020, Thermo Fisher Scientific), in accordance with the manufacturer's instructions. The yielded plasmid (pENTR-hMPI) was used as a template to amplify the *MPI* gene by PCR using a primer set (*Supplementary file 1*) and the PCR products were cloned into pMXs-Neo retroviral expression vector (RTV-011, Cell Biolabs) to yield pMXs-Neo-hMPI by using In-Fusion HD Cloning Kit (639648, Clontech), in accordance with the manufacturer's instructions. mCherry-hCdt1(1/100)Cy(−)/pcDNA3 was purchased from RIKEN BioResource Research Center. The coding sequence of mCherry-hCdt1(1/100)Cy(−) was amplified by PCR using a primer set (*Supplementary file 1*) and cloned into pMXs-Neo to yield pMXs-Neo-mCherry-hCdt1(1/100)Cy(−) using the In-Fusion HD Cloning Kit. pMMLV-mVenus-hGem(1/110):IRES:Bsd was constructed and purchased from Vector Builder.

## Retroviral packaging and transduction

The Plat-A retroviral packaging cell line (1 × 10⁶ cells, Cell Biolabs) was seeded on collagen I-coated six-well plates (4810–-010, IWAKI) and cultured in complete medium for 24 hr. After the culture medium had been replaced with 1 mL of fresh complete medium, 1 mL of Opti-MEM containing 3 µg of plasmids (*Supplementary file 2*), 9 µL of Lipofectamine 3000, and 6 µL of P3000 reagent (L3000015, Thermo Fisher Scientific) was added to the cells, followed by culture for 48 hr. The culture medium was then passed through a 0.80 µm syringe filter (SLAA033SS, Merck Millipore). The filtrate, which contained retroviral particles, was mixed with hexadimethrine bromide (8 µg/mL; 17736-44, Nacalai Tesque) and Man (20 µM), and the retroviral solution (1 mL) was added to MPI-KO HT1080 cells or MPI-KO HeLa cells (5 × 10⁴ cells/six-well plates) that were cultured for 24 hr in complete medium supplemented with 20 µM Man. The transduced cells were selected for 14 d in complete medium supplemented with [for pMXs-Neo, pMXs-Neo-mCherry-hCdt1(1/100)Cy(−), and pMMLV-mVenus-hGem(1/110):IRES:Bsd] or without (pMXs-Neo-hMPI) 20 µM Man, and 600 µg/mL G418 [09380-44, Nacalai Tesque; for pMXs-Neo, pMXs-Neo-hMPI, and pMXs-Neo-mCherry-hCdt1(1/100)Cy(−)] or 13 µg/mL blasticidin S [A1113903, Thermo Fisher Scientific; for pMMLV-mVenus-hGem(1/110):IRES:Bsd].

## Real-time polymerase chain reaction

Real-time polymerase chain reaction (PCR) was performed with the 7500 Real-Time PCR System (Applied Biosystems). cDNA (0.5 µL) was mixed in 20 µL of a reaction mixture that contained THUNDERBIRD Next SYBR qPCR Mix (10 µL, QPX-201, TOYOBO) and a primer set (0.06 µL each, *Supplementary file 1*). cDNA was amplified by an initial denaturation step at 95°C for 30 s, followed by 40 cycles of 95°C for 5 s and 55°C for 35 s. The samples were analyzed in duplicate, and the mean number of cycles required to reach the threshold level of fluorescent detection was calculated for each sample. *ACTB* expression was used to normalize the amounts of cDNA in each sample.

## Preparation of whole-cell lysates

Cells were rinsed once with PBS, scraped in the same buffer, and pelleted by centrifugation at 200 × *g* for 5 min at 4°C. The wet cell weight was measured after the supernatant had been completely removed. The cell pellet was resuspended at a concentration of 10 mg wet cells/100 µL of PBS. The cell suspension was mixed with an equal volume of 2× Laemmli sample buffer containing 10% β-mercaptoethanol, 1× cOmplete Protease Inhibitor Cocktail (11836170001, Sigma-Aldrich), and 1× PhosSTOP (4906845001, Sigma-Aldrich), and homogenized by two cycles of 10 s sonication at 0°C using a probe-type sonicator (VP-5S, TAITEC). The cell homogenates were denatured by heating at 100°C for 3 min and stored at −80°C until use.

## Chromatin extraction

Cells were washed once with PBS, scraped in the same buffer, and pelleted by centrifugation at 200 × *g* for 5 min at 4°C. The wet cell weight was measured after the supernatant had been completely removed. The cell pellet was extracted by incubating for 5 min at 0°C at a concentration of 10 mg wet cells/100 µL of CSK buffer (10 mM HEPES-KOH, pH 7.4, 340 mM sucrose, 10% [v/v] glycerol, 10 mM KCl, 1.5 mM $MgCl_2$, 0.1% [v/v] Triton X-100, 1 mM ATP, 1 mM DTT, 1× cOmplete Protease Inhibitor Cocktail, and 1× PhosSTOP). The homogenate was centrifuged at 1390 × *g* for 5 min at 4°C and the pellet was resuspended in 100 µL of CSK buffer. The homogenate was centrifuged again at 1390 × *g* for 5 min at 4°C. The pellet was resuspended in 100 µL of CSK buffer and mixed with 100 µL of 2× Laemmli sample buffer containing 10% β-mercaptoethanol. The samples were homogenized by two cycles of 10 s sonication at 0°C using a probe-type sonicator (VP-5S, TAITEC) and the homogenates were denatured by heating at 100°C for 3 min.

## Western and lectin blotting

Whole-cell lysates or chromatin fractions (250 µg of wet cells) were separated by sodium dodecyl sulfate (SDS)-polyacrylamide gel electrophoresis (PAGE) and analyzed by lectin blotting using biotinylated concanavalin A (Con A) lectin (1:10,000, J203, Cosmo Bio) and VECTASTAIN ABC-HRP kit (PK-4000, Vector Laboratories) or by western blotting using primary antibodies and secondary antibodies. The primary antibodies were as follows: anti-MPI (1:5000, GTX103682, GeneTex), anti-β-actin (1:10,000, 010-27841, FUJIFILM Wako), anti-H2AX (1:2000, 938CT5.1.1, Santa Cruz), anti-phospho-H2AX (Ser139) (γH2AX; 1:5000, JBW301, Millipore), anti-RPA2 (1:5000, 9H8, Santa Cruz), anti-Phospho-RPA2 (S33) (1:10,000, A300-246A, Bethyl), anti-MCM2 (1:10,000, D7G11, Cell Signaling Technology), anti-MCM3 (1:10,000, E-8, Santa Cruz), anti-MCM4 (1:5000, GTX109740, GeneTex), anti-MCM5 (1:5000, GTX33310, GeneTex), anti-MCM6 (1:20,000, H-8, Santa Cruz), anti-MCM7 (1:5000, 141.2, Santa Cruz), anti-CDT1 (1:10,000, ab202067, Abcam), anti-CDC6 (1:1000, ab109315, Abcam), anti-ORC2 (1:10,000, 3G6, Santa Cruz), and anti-CDC45 (1:1000, D7G6, Cell Signaling Technology). The secondary antibodies were as follows: horseradish peroxidase (HRP)-conjugated goat anti-mouse immunoglobulin (IgG) (1:10,000, 7076, Cell Signaling Technology), HRP-conjugated goat anti-rabbit IgG (1:10,000, 7074, Cell Signaling Technology), and HRP-conjugated goat anti-rat IgG (1:5000, 7077, Cell Signaling Technology).

## Cell cycle analysis

Cells were labeled with 10 µM BrdU (B1575, Tokyo Chemical Industry) for 1 hr before harvest. Trypsinized cells (1 × $10^6$ cells) were washed two times with ice-cold PBS and fixed in 1 mL of 70% ethanol for no less than 16 hr at 4°C. After washing two times with PBS, fixed cells were denatured by incubating in 500 µL of 2.0 M HCl for 30 min at 25°C. Denatured cells were washed two times with PBS

and once with 1 mL of sodium borate buffer, pH 8.5. The cells were permeabilized in 500 µL of PBS containing 0.1% Triton X-100 (PBSTx) for 10 min at 25°C. The permeabilized cells were incubated for 1 hr at 25°C in 500 µL of PBS containing 1% bovine serum albumin (BSA), Alexa Fluor 488-conjugated anti-BrdU antibody (1:100, 3D4, BioLegend), and Cellstain Hoechst 33342 solution (1:100, H342, Dojindo). The cells were then washed once with 1 mL of PBS containing 1% BSA (PBS/BSA) and resuspended in 500 µL of the same buffer. The stained samples were analyzed by BD LSRFortessa X-20 and FlowJo v10.6 (Becton Dickinson).

## Chromatin flow cytometry

Trypsinized cells ($1 \times 10^6$ cells) were extracted by incubating for 5 min at 0°C in 500 µL of CSK buffer. The cells were washed by adding 1 mL of ice-cold PBS/BSA, pelleted by centrifugation at $1390 \times g$ for 5 min at 4°C, and fixed for 15 min at 25°C in 500 µL of 4% paraformaldehyde (161-20141, FUJIFILM Wako). After quenching the reaction by adding 1 mL of PBS/BSA, the cells were pelleted by centrifugation at $2000 \times g$ for 7 min at 4°C and washed again with 1 mL of PBS/BSA. The washed cells were then resuspended in 200 µL of PBS containing 1% BSA and 0.1% Triton X-100 (PBSTx/BSA) containing primary antibodies (MCM2 [1:200, D7G11, Cell Signaling Technology] and γH2AX [1:200, JBW301, Millipore]) and incubated for 1 hr at 25°C. After washing the cells once in 1 mL of PBSTx/BSA, they were incubated for 1 hr at 25°C in the dark with 200 µL of PBSTx/BSA containing secondary antibodies (R37118, donkey anti-rabbit IgG-Alexa Fluor 488 [1:1,000, Thermo Fisher Scientific]; A-31571, donkey anti-mouse IgG-Alexa Fluor 647 [1:1000, Thermo Fisher Scientific]) and Cellstain Hoechst 33342 solution (1:200). The cells were washed once in 1 mL of PBSTx/BSA, resuspended in 500 µL of the same buffer, and subjected to flow cytometry.

## Cell viability (ATP) assay

Cells ($1 \times 10^3$ cells/96-well plate) were seeded and cultured for 24 hr in 100 µL of complete medium supplemented with 20 µM Man. For co-treatment assay, the cells were incubated with drugs or vehicle alone for 24 hr in complete medium supplemented with 50 µM or 5 mM Man. For a preconditioning assay, cells were incubated for 24 hr in complete medium supplemented with 50 µM or 5 mM Man, and the preconditioned cells were further incubated with drugs or vehicle alone for 24 hr. After the drug treatment, CellTiter-Glo 2.0 (100 µL/well, G924B, Promega) was added to the cells and incubated for 10 min at 25°C before measuring luminescence with an integration time of 1000 ms using an Infinite 200 Pro microplate reader (Tecan).

## LDH assay

For *Figure 5E*, cells ($1 \times 10^3$ cells/96-well plate) that were cultured for 24 hr in 100 µL of complete medium supplemented with 20 µM Man were treated with 30 nM IACS-010759 or dimethylsulfoxide (DMSO) alone for an additional 6 hr in complete medium supplemented with unchallenged or challenged concentrations of Man. Total LDH activity was measured using Cytotoxicity LDH Assay Kit-WST (CK12, Dojindo), in accordance with the manufacturer's instructions. For *Figure 5F* (necrosis assay), cells ($3.3 \times 10^4$ cells/24-well plate) that were cultured for 24 hr in 500 µL of complete medium supplemented with 20 µM Man were treated with 30 nM IACS-010759 or DMSO alone for an additional 24 hr in 500 µL of complete medium supplemented with unchallenged or challenged concentrations of Man. After centrifugation at $250 \times g$ for 5 min at 4°C, the culture supernatant (50 µL, extracellular LDH) was transferred to a 96-well plate. The cells were solubilized by adding 20 µL of 10% (w/v) Tween 20 for 30 min at 37°C with gentle agitation. The homogenates (50 µL, total LDH) were transferred to the same 96-well plate prepared as above. LDH activity was measured using Cytotoxicity LDH Assay Kit-WST, in accordance with the manufacturer's instructions.

## Seahorse real-time cell metabolic analysis

The ATP rate assay was performed using an XFe96 Extracellular Flux analyzer, in accordance with the manufacturer's instructions (Seahorse Bioscience). MPI-KO HT1080 cells (#3, $1 \times 10^4$ cells/96-well assay plate) were seeded and incubated for 18 hr in complete DMEM. Prior to the assay, the cells were preincubated for 60 min at 37°C in XF DMEM medium containing 10% FBS, 10 mM glucose, 4 mM L-glutamine, and 50 µM Man. The cells were then treated with 5 mM Man or medium alone for 120 min, followed by incubation with 1.5 µM oligomycin for 18 min and then 0.5 µM each of rotenone

and antimycin A for 18 min. Data analysis was performed using the XF Real-Time ATP Rate Assay Report Generator (Seahorse Bioscience).

## Label-free proteomics

Cells ($5 \times 10^5$ cells/10 cm dish) were cultured for 24 hr in 10 mL of complete medium supplemented with 20 µM Man. The cells were further incubated for 1, 2, and 6 d in 10 mL of complete medium supplemented with 50 µM Man (1 and 2 d) or 5 mM Man (1, 2, and 6 d). For 6-day incubation, the medium was replaced every other day. The cells were washed two times with PBS and scraped in 1 mL of PBS. After the cells had been pelleted by centrifugation at $200 \times g$ for 5 min at 4°C, the cell pellets were flash-frozen in liquid nitrogen and stored at −80°C until use. Protein precipitates were obtained by adding 10% trichloroacetic acid to freeze-thawed cells and centrifuging at $12,000 \times g$ for 20 min. After washing the precipitates three times with acetone, they were dissolved with 7 M guanidine, 1 M Tris-HCl (pH 8.5), 10 mM EDTA, and 50 mM dithiothreitol. After alkylation with iodo-acetic acid, samples were desalted using PAGE Clean Up Kit (06441-50, Nacalai Tesque). The resultant precipitates were dissolved with 20 mM Tris-HCl (pH 8.0), 0.03% (w/v) n-dodecyl-β-D-maltoside, and digested with trypsin (tosyl phenylalanyl chloromethyl ketone-treated; Worthington Biochemical) at 37°C for 18 hr. The concentration of the peptide mixture was quantified by amino acid analysis (*Masuda and Dohmae, 2011*). 1 µg of each peptide mixture was subjected to liquid chromatography (LC)-tandem mass spectrometry (MS/MS). Solvent A (0.1% formic acid) and solvent B (80% acetonitrile with 0.1% formic acid) were used as eluents. Peptides were separated using an Easy nLC 1200 (Thermo Fisher Scientific) equipped with a nano-ESI spray column (NTCC-360, 0.075 mm internal diameter × 105 mm length, 3 µm, Nikkyo Technos) at a flow rate of 300 nL/min under linear gradient conditions over 250 min. The separated peptides were analyzed with an online coupled Q Exactive HF-X Mass Spectrometer (Thermo Fisher Scientific) using the data-dependent Top 10 method. The acquired data were processed using MASCOT 2.8 (Matrix Science) and Proteome Discoverer 2.4 (Thermo Fisher Scientific). The MASCOT search was conducted as follows: Database, NCBIprot; taxonomy, *Homo sapiens* (human) (438,061 sequences); type of search, MS/MS ion; enzyme, trypsin; fixed modification, none; variable modifications, acetyl (protein N-term), Gln->pyro-Glu (N-term Q), oxidation (M), carboxymethyl (C); mass values, monoisotopic; peptide mass tolerance, ±15 ppm; fragment mass tolerance,±30 mmu; max. missed cleavages, 3; and instrument type, ESI-TRAP. Label-free quantification was performed using the quantification method based on the ion intensity of peptides in Proteome Discoverer 2.4.

## Metabolomics

Approximately $8 \times 10^6$ cells were washed two times with 5 mL of 5% (w/v) mannitol (133-00845, FUJI-FILM Wako) and scraped in 1.3 mL of methanol (138-14521, FUJIFILM Wako) that was spiked with 10 µM external standards (Human Metabolome Technologies). The cell homogenates were spun at $15,000 \times g$ for 5 min at 4°C, and the wet cell weights were measured. The supernatant was analyzed by capillary electrophoresis (CE) time-of-flight (TOF) mass spectrometry (MS) using an Agilent CE-TOFMS system (Agilent Technologies) at Human Metabolome Technologies.

## [$^{13}C_6$]-Glucose-tracing experiments

Approximately $6 \times 10^6$ cells were washed two times with 5 mL of PBS and incubated in 10 mL of glucose-free DMEM (042-32255, FUJIFILM Wako) containing 5 mM [$^{13}C_6$]-glucose (CLM-1396-1, Cambridge Isotope Laboratories), 50 µM or 5 mM Man, and 10% dialyzed FBS (SH30079.02, Cytiva) for the indicated time. The labeled cells were harvested at the indicated time as described in 'Metabolomics.' These samples were analyzed using capillary ion chromatography-mass spectrometry and liquid chromatography-mass spectrometry as previously described (*Hirayama et al., 2020*; *Suzuki et al., 2022*).

## Time-lapse imaging

Fucci MPI-KO HT1080 cells (#3 subclone 9-5, $4 \times 10^4$ cells/six-well dish) were cultured for 24 hr in 2.5 mL of complete medium supplemented with 20 µM Man. The cells were washed once with complete DMEM with no phenol red (040-30095, FUJIFILM Wako), 4 mM glutamine, and 10% FBS. The washed cells were cultured in 2.5 mL of the same medium supplemented with 50 µM Man or

5 mM Man in a humidified chamber (Tokai Hit) at 37°C with 5% $CO_2$. Fluorescence and differential interference contrast images were obtained every 15 min using KEYENCE BZ-X800 with a PlanFluor ×20 objective lens (NA = 0.45, WD = 8.80–7.50 mm, Ph1; KEYENCE) or EVIDENT FLUOVIEW FV10i. In KEYENCE BZ-X810, Fucci mCherry and mVenus signals were detected using a TRITC filter (Ex, 545/25 nm; Em, 605/70 nm; KEYENCE) and mVenus filter (Ex, 500/20; Em, 535/30 nm; M SQUARE), respectively. In EVIDENT FLUOVIEW FV10i, Fucci mCherry and mVenus signals were detected using mCherry (Ex, 559; Em, 570-620) and EYFP (Ex, 473; Em, 490-540) dye settings. Images were acquired after a 30 min equilibration period.

## Quantification

For western blots, the X-ray films were scanned using an EPSON GT-7600UF scanner. ImageJ (*Schneider et al., 2012*) was used for the quantification of band intensity. For image processing of Fucci time-lapse imaging, a Fiji/ImageJ (*Schindelin et al., 2012*) plugin, Trackmate (*Tinevez et al., 2017*), was used to track single cells. MATLAB (MathWorks, Natick, MA) was used to quantify the duration of each cell cycle phase. The Fucci profiles in the mannose-challenged cells (*Figure 2E*, Figure S4) were visually inspected for classification.

## Statistical analysis

R version 3.3.3 and Prism 9 were used for statistical analysis. Statistical analysis was performed by applying an unpaired two-sided Welch's *t*-test for comparison of the means between two groups. Comparisons of the means among more than two groups were performed with one-way or two-way analysis of variance (ANOVA) followed by *post hoc* testing with Dunnett's test, Tukey's or Bonferroni's test. Data are reported as the mean ± standard deviation (SD). p-Values<0.05 were considered to be statistically significant.

## Acknowledgements

We wish to thank our laboratory members, Drs. Takashi Akazawa and Yosuke Matsuoka (Osaka International Cancer Institute), for fruitful discussions, Dr. Steven A Rosenberg for MCA205 cells, and Drs. Kazuki Nakajima (Gifu University), Takuro Tojima (RIKEN), Michiko Kodama (Osaka University) and Natsuki Osaka (Keio University) for technical advice and support. We thank Edanz (https://jp.edanz.com/ac) for editing a draft of this manuscript. This study was funded by The Takeda Science Foundation (YH), JSPS KAKENHI Grant Number JP23K06645 (YH), The Rocket Fund (HHF), and R01DK99551 (HHF).

## Additional information

### Funding

| Funder | Grant reference number | Author |
|---|---|---|
| The Takeda Science Foundation | | Yoichiro Harada |
| The Rocket Fund | | Hudson H Freeze |
| National Institutes of Health | R01DK99551 | Hudson H Freeze |
| KAKENHI | JP23K06645 | Yoichiro Harada |

The funders had no role in study design, data collection and interpretation, or the decision to submit the work for publication.

### Author contributions

Yoichiro Harada, Conceptualization, Resources, Data curation, Formal analysis, Funding acquisition, Validation, Investigation, Visualization, Methodology, Writing - original draft, Writing - review and editing; Yu Mizote, Takehiro Suzuki, Mikako Nishida, Yusuke Imagawa, Data curation, Formal analysis, Investigation, Writing - original draft, Writing - review and editing; Akiyoshi Hirayama, Kento Maeda,

Data curation, Formal analysis, Investigation, Writing - review and editing; Satsuki Ikeda, Investigation; Toru Hiratsuka, Software, Formal analysis, Writing - original draft, Writing - review and editing; Ayaka Ueda, Data curation, Investigation, Writing - review and editing; Yuki Ohkawa, Data curation, Writing - review and editing; Junko Murai, Eiji Miyoshi, Shigeki Higashiyama, Heiichiro Udono, Hideaki Tahara, Naoyuki Taniguchi, Supervision, Writing - review and editing; Hudson H Freeze, Resources, Writing - review and editing; Naoshi Dohmae, Data curation, Formal analysis, Supervision, Writing - review and editing

### Author ORCIDs
Yoichiro Harada http://orcid.org/0000-0003-1818-9633
Naoshi Dohmae http://orcid.org/0000-0002-5242-9410

### Decision letter and Author response
Decision letter https://doi.org/10.7554/eLife.83870.sa1
Author response https://doi.org/10.7554/eLife.83870.sa2

---

## Additional files

### Supplementary files
• Supplementary file 1. List of primers used in this study.
• Supplementary file 2. List of plasmids used in this study.
• MDAR checklist
• Source data 1. Proteomic data.
• Source data 2. Functional annotation analysis of proteomic data.
• Source data 3. Metabolomic data.
• Source data 4. Metabolomic data for stable isotope labeling experiments.

### Data availability
The mass spectrometry proteomics data have been deposited to the ProteomeXchange Consortium via the PRIDE partner repository with the dataset identifier PXD036449 and https://doi.org/10.6019/PXD036449.

The following dataset was generated:

| Author(s) | Year | Dataset title | Dataset URL | Database and Identifier |
|-----------|------|---------------|-------------|-------------------------|
| Suzuki T, Dohmae N | 2022 | Metabolic clogging of mannose in human cancer cells triggers genome instability via dNTP loss | https://www.ebi.ac.uk/pride/archive/projects/PXD036449 | PRIDE, PXD036449 |

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

# Appendix 1

## Appendix 1—key resources table

| Reagent type (species) or resource | Designation | Source or reference | Identifiers | Additional information |
|---|---|---|---|---|
| Gene (*Homo sapiens*) | MPI | GenBank | Gene ID: 4351 | |
| Genetic reagent (*H. sapiens*) | MPI-KO HT1080 | This paper | | Clone #2, #3, which can be obtained from the Department of Glyco-Oncology and Medical Biochemistry, Osaka International Cancer Institute |
| Genetic reagent (*H. sapiens*) | Fucci MPI-KO HT1080 | This paper | | Clone #3 subclone (9-5), which can be obtained from the Department of Glyco-Oncology and Medical Biochemistry, Osaka International Cancer Institute |
| Genetic reagent (*H. sapiens*) | MPI-KO HeLa | This paper | | Clone #1, #2, which can be obtained from the Department of Glyco-Oncology and Medical Biochemistry, Osaka International Cancer Institute |
| Genetic reagent (*Mus musculus*) | Wild-type mouse embryonic fibroblasts | *DeRossi et al., 2006* | | |
| Genetic reagent (*M. musculus*) | MPI-KO mouse embryonic fibroblasts | *DeRossi et al., 2006* | | |
| Cell line (*H. sapiens*) | HT1080 | Our laboratory stock | | Validated by short tandem repeat profiling |
| Cell line (*H. sapiens*) | HeLa | RIKEN BioResource Research Center | RCB0007 | |
| Cell line (*H. sapiens*) | KP-4 | RIKEN BioResource Research Center | RCB1005 | |
| Cell line (*H. sapiens*) | SK-OV-3 | American Type Culture Collection (ATCC) | HTB-77 | |
| Cell line (*H. sapiens*) | A549 | ATCC | CRM-CCL-185 | |
| Cell line (*M. musculus*) | MCA205 | A gift from Rosenberg lab | | |
| Cell line (*H. sapiens*) | The Plat-A retroviral packaging cell line | Cell Biolabs | RV-102 | |
| Antibody | Anti-MPI (rabbit polyclonal) | GeneTex | GTX103682 | WB (1:5000) |
| Antibody | Anti-β-actin (mouse monoclonal) | FUJIFILM Wako | 010-27841 | WB (1:10,000) |
| Antibody | Anti-H2AX (mouse monoclonal) | Santa Cruz | 938CT5.1.1 | WB (1:2000) |
| Antibody | Anti-phospho-H2AX (Ser139) (mouse monoclonal) | Millipore | JBW301 | WB (1:5000) Chromatin FACS (1:200) |
| Antibody | Anti-RPA2 (mouse monoclonal) | Santa Cruz | 9H8 | WB (1:5000) |
| Antibody | Anti-Phospho-RPA2 (S33) (rabbit polyclonal) | Bethyl | A300-246A | WB (1:10,000) |
| Antibody | Anti-MCM2 (rabbit monoclonal) | Cell Signaling Technology | D7G11 | WB (1:10,000) Chromatin FACS (1:200) |
| Antibody | Anti-MCM3 (mouse monoclonal) | Santa Cruz | E-8 | WB (1:10,000) |
| Antibody | Anti-MCM4 (rabbit polyclonal) | GeneTex | GTX109740 | WB (1:5000) |
| Antibody | Anti-MCM5 (rabbit polyclonal) | GeneTex | GTX33310 | WB (1:5000) |
| Antibody | Anti-MCM6 (mousemonoclonal) | Santa Cruz | H-8 | WB (1:20,000) |
| Antibody | Anti-MCM7 (mouse monoclonal) | Santa Cruz | 141.2 | WB (1:5000) |
| Antibody | Anti-CDT1 (rabbit monoclonal) | Abcam | ab202067 | WB (1:10,000) |
| Antibody | Anti-CDC6 (rabbit monoclonal) | Abcam | ab109315 | WB (1:1000) |
| Antibody | Anti-ORC2 (rat monoclonal) | Santa Cruz | 3G6 | WB (1:10,000) |
| Antibody | Anti-CDC45 (rabbit monoclonal) | Cell Signaling Technology | D7G6 | WB (1:10,000) |
| Antibody | Alexa Fluor 488-conjugated anti-BrdU antibody (mouse monoclonal) | BioLegend | 3D4 | FACS (1:100) |

*Appendix 1 Continued on next page*

*Appendix 1 Continued*

| Reagent type (species) or resource | Designation | Source or reference | Identifiers | Additional information |
|---|---|---|---|---|
| Antibody | Anti-rabbit IgG-Alexa Fluor 488 (donkey polyclonal) | Thermo Fisher Scientific | R37118 | Chromatin FACS (1:1000) |
| Antibody | Anti-mouse IgG-Alexa Fluor 647 (donkey polyclonal) | Thermo Fisher Scientific | A-31571 | Chromatin FACS (1:1000) |
| Recombinant DNA reagent | pENTR-hMPI | This paper | | This reagent can be obtained from the Department of Glyco-Oncology and Medical Biochemistry, Osaka International Cancer Institute |
| Recombinant DNA reagent | pMXs-Neo-hMPI | This paper | | This reagent can be obtained from the Department of Cancer Drug Discovery and Development, Osaka International Cancer Institute |
| Recombinant DNA reagent | mCherry-hCdt1(1/100)Cy(−)/pcDNA3 | RIKEN BioResource Research Center | RDB15459 | |
| Recombinant DNA reagent | pMXs-Neo to yield pMXs-Neo-mCherry-hCdt1(1/100)Cy(−) | This paper | | This reagent can be obtained from the Department of Cancer Drug Discovery and Development, Osaka International Cancer Institute |
| Recombinant DNA reagent | pMMLV-mVenus-hGem(1/110):IRES:Bsd | Vector Builder | | |
| Recombinant DNA reagent | Edit-R SMART Cas9_mCMV_(PuroR) expression plasmid | Dharmacon | U-005200-120 | |
| Recombinant DNA reagent | pENTR/D-TOPO vector | Thermo Fisher Scientific | K240020 | |
| Recombinant DNA reagent | pMXs-Neo retroviral expression vector | Cell Biolabs | RTV-011 | |
| Sequence-based reagent | Synthetic CRISPR RNAs for human *MPI* | Dharmacon | CM-011729-01 CM-011729-03 | |
| Sequence-based reagent | trans-activating CRISPR RNA | Dharmacon | U-002005-05 | |
| Commercial assay or kit | VECTASTAIN ABC-HRP kit | Vector Laboratories | PK-4000 | |
| Commercial assay or kit | CellTiter-Glo 2.0 | Promega | G924B | |
| Commercial assay or kit | Cytotoxicity LDH Assay Kit-WST | Dojindo | CK12 | |
| Commercial assay or kit | PAGE Clean Up Kit | Nacalai | 06441-50 | |
| Commercial assay or kit | In-Fusion HD Cloning Kit | Clontech | 639648 | |
| Commercial assay or kit | RNeasy Mini Kit | QIAGEN | 74106 | |
| Commercial assay or kit | NucleoSpin Gel and PCR Clean-Up Kit | Clontech | 740609 | |
| Commercial assay or kit | THUNDERBIRD Next SYBR qPCR Mix | TOYOBO | QPX-201 | |
| Chemical compound, drug | Cellstain Hoechst 33342 solution | Dojindo | H342 | FACS (1:100) Chromatin FACS (1:200) |
| Chemical compound, drug | 5-Bromo-2'-deoxyuridine | Tokyo Chemical Industry | B1575 | |
| Chemical compound, drug | 2'-Deoxyadenosine | Wako | 046-18693 | |
| Chemical compound, drug | Thymidine | Wako | 203-19423 | |
| Chemical compound, drug | 2'-Deoxyguanosine | Tokyo Chemical Industry | D0052 | |
| Chemical compound, drug | 2'-Deoxycytidine | Tokyo Chemical Industry | D3583 | |
| Chemical compound, drug | Dialyzed FBS | Cytiva | SH30079.02 | |

*Appendix 1 Continued*

| Reagent type (species) or resource | Designation | Source or reference | Identifiers | Additional information |
|---|---|---|---|---|
| Chemical compound, drug | Cisplatin | Tokyo Chemical Industry | D3371 | |
| Chemical compound, drug | Doxorubicin | FUJIFILM Wako | 040-21521 | |
| Chemical compound, drug | IACS-010759 | Selleckchem | S8731 | |
| Chemical compound, drug | VE-821 | Sigma-Aldrich | SML1415 | |
| Chemical compound, drug | Hydroxyurea | FUJIFILM Wako | 085-06653 | |
| Chemical compound, drug | [$^{13}C_6$]-Glucose | Cambridge Isotope Laboratories | CLM-1396-1 | |
| Chemical compound, drug | cOmplete Protease Inhibitor Cocktail | Sigma-Aldrich | 11836170001 | |
| Chemical compound, drug | PhosSTOP | Sigma-Aldrich | 4906845001 | |
| Chemical compound, drug | Biotinylated concanavalin A | Cosmo Bio | J203 | Lectin blot (1:10,000) |
| Chemical compound, drug | TRIzol Reagent | Thermo Fisher Scientific | 15596026 | |
| Chemical compound, drug | Lipofectamine 3000 | Thermo Fisher Scientific | L3000015 | |
| Chemical compound, drug | Lipofectamine 2000 | Thermo Fisher Scientific | 11668027 | |
| Software, algorithm | MASCOT 2.8 | Matrix Science | | |
| Software, algorithm | Proteome Discoverer 2.4 | Thermo Fisher Scientific | | |
| Software, algorithm | ImageJ | *Schneider et al., 2012* | | |
| Software, algorithm | Trackmate | *Tinevez et al., 2017* | | |
| Software, algorithm | MATLAB | MathWorks | | |

