## [Editor Report]

Mannose is toxic to honeybees and some cancer cells that lack sufficient capacity to metabolize this sugar. However, the precise metabolic consequences of impaired mannose metabolism require further understanding. In this important study, Harada et al. provide convincing evidence that an inability to metabolize mannose leads to impaired synthesis of deoxynucleotide triphosphates and DNA, which can be leveraged to sensitize cancer cells to chemotherapy.

---

## [Decision Letter]

**Decision letter after peer review:**

Thank you for submitting your article "Metabolic clogging of mannose triggers genomic instability via dNTP loss in human cancer cells" for consideration by *eLife*. Your article has been reviewed by 3 peer reviewers, one of whom is a member of our Board of Reviewing Editors, and the evaluation has been overseen by Richard White as the Senior Editor. The following individual involved in review of your submission has agreed to reveal their identity: John A Hanover (Reviewer #3).

Essential revisions:

1) The authors should extend their observations to a larger panel of cell lines, including MPI high and low cell lines.

2) The authors should perform stable isotope tracing experiments to examine the activity of key metabolic pathways (e.g. the pentose phosphate pathway) instead of inferring activity from metabolite pool sizes.

3) The authors should further develop their metabolic characterization of the consequences of the inability to metabolize mannose, including the effects on the TCA cycle and oxidative phosphorylation and dNTP depletion as requested by reviewers #1 and #2.

4) The authors should demonstrate causality for dNTP depletion in the reduction of proliferation and cell cycle perturbation.

5) The authors should improve data presentation, including showing individual data points and improving the description of the variables as suggested by the reviewers.

6) When giving context to their work, the authors should more thoroughly discuss prior work on cancer cells (not just honeybees), including very in-depth metabolic phenotyping of mannose phosphate isomerase low and high cells that was recently reported.

*Reviewer #1 (Recommendations for the authors):*

While the manuscript is well-written and advances our understanding of mannose toxicity based on new findings that dNTP depletion occurs, several concerns remain that could be addressed by the authors:

1. The metabolic findings from Gonzalez et al. are insufficiently discussed in the text. Their study already established, using sophisticated stable isotope tracing approaches, that mannose decreases the contribution of glucose to G6P and lactate, and results in lower levels of glycolytic and TCA cycle intermediates, and the PPP intermediate ribose-5P. Although the authors cite this study in the context of suppression of proliferation and increased chemotherapeutic efficacy, what is already known about mannose metabolism in mammalian cells is understated in the manuscript and the focus on honeybees is overemphasized.

2. Gonzalez et al., found that mannose decreased ribose-5P but in this study no difference in ribose-5P was seen upon mannose treatment. Total measurement of PPP metabolites is not indicative of pathway flux and the authors should instead use 1,2-13C-glucose tracing to determine whether mannose influences PPP flux. Moreover, U-13C-glucose can be used to look at dNTP labeling.

3. The mechanistic connection between mannose metabolism and dNTP depletion is not established. If there is no effect of mannose on the PPP as claimed, how does mannose induce dNTP depletion to induce replication stress?

4. It is unclear whether dNTP depletion is a cause or consequence of reduced proliferation and cell cycle perturbation. Direct evidence is lacking. Can nucleosides rescue mannose toxicity, cell cycle progression, chemosensitivity and DNA synthesis?

*Reviewer #2 (Recommendations for the authors):*

1. Identification of cancer cell lines that may be susceptible to high mannose-induced toxicity, such as those with endogenous low expression levels of MPI, and demonstration of the same effects in those cell lines would strengthen the study.

2. Utilization of labeled isotope tracers such as 13C-glucose, 13C-mannose, or 13C-glutamine to trace metabolic flux would provide confidence in the conclusions about the changes to the metabolic landscape. Pool sizes decreased for both lower glycolysis and TCA cycle intermediates; this was interpreted as decreased activity for glycolysis but increased activity for the TCA cycle. How is the TCA cycle fueled if lower glycolytic intermediates are not feeding the TCA cycle? Is excess glutamine fueling the TCA cycle? Why are nucleoside monophosphates not depleted?

3. Evaluating the NAD/NADH ratio between mannose challenged and unchallenged cells would also strengthen the claims about increased oxidative phosphorylation.

4. For Seahorse experiments, data following oligomycin, rotenone, and antimycin A addition should be included.

5. Figure 1G: would removal of mannose further reduce hexose-6-phosphate levels?

6. When cell growth is restored after re-introduction of physiologic mannose as shown in Figure 2G, will these cells revert back to a normal cell cycle progression as shown in Figure 2B, or do they retain aberrant cell cycle progression patterns?

7. Figure 2I and J do not show BRdU incorporation into MPI-KO HT1080 cells under 50 μM mannose on day 6. Similarly, Figure 2H does not include p21 and p27 expression for day 6 in the 50 μM mannose condition. Please justify.

8. Inclusion of individual data points for each replicate within bar graphs would improve the reader's ability to interpret results.

*Reviewer #3 (Recommendations for the authors):*

Specific Points:

The data in Figure 1. The induction of honeybee syndrome suppresses cell proliferation and increases chemosensitivity. Shows only modest changes in chemosensitivity and the discussion does not adequately express this. The authors should further discuss this.

Figure 3. These results are discussed almost exclusively in the context of DNA replication and cell cycle for the down regulated pathways. What about ribosomes?

The data are clear but it is hard to assess the variability in some of the assays since there is scant description of the variables particularly the cell sorting experiments to establish DNA replication and cell cycle.

[Editors’ note: further revisions were suggested prior to acceptance, as described below.]

Thank you for resubmitting your work entitled "Metabolic clogging of mannose triggers dNTP loss and genomic instability in human cancer cells" for further consideration by *eLife*. Your revised article has been evaluated by Richard White (Senior Editor) and a Reviewing Editor.

The manuscript has been improved but there are some remaining issues that need to be addressed, as outlined below:

The stable isotope tracing experiments have problems that leave this point unaddressed. Specifically:

1. It is very difficult to distinguish the isotopologues because of the colors used and the legend, which has too many possibilities beyond the masses possible. The authors should simplify these figures and use colors that are more easily distinguishable.

2. U-13C6-glucose is not suitable to assay the non-oxidative and oxidative branches of the PPP and thus the authors cannot conclude that mannose decreases flux into this pathway. The authors should use 1,2-C2-glucose tracing to draw these conclusions and also perform labeling with additional tracers (e.g. serine, glutamine, mannose) to parse where the defects in nucleotide synthesis are occurring.

3. The authors should address other issues raised in the comments below.

*Reviewer #1 (Recommendations for the authors):*

The authors have done a good job addressing concerns about (1) extending their observations to a larger panel of MPI high and low cell lines, (2) further develop their characterization of the effects of mannose on the TCA cycle and dNTPs, and (3) examine the causality for dNTP depletion in genomic instability. However, the stable isotope tracing experiments have problems that leave this point unaddressed. The authors chose U-13C6-glucose instead of 1,2-C2-glucose, and this is suitable to assay the TCA cycle. However, it is unsuitable to assay the non-oxidative and oxidative branches of the PPP because these rapidly saturate and labeling in R5P from either branch cannot be distinguished. This is very evident in Figure 6—figure supplement 4, where both 6PG and S7P are completely labeled at all time points and both conditions, with only the total levels different. It is unclear why mannose decreases the fraction labeling in Ru5P + R5P specifically. Therefore, the authors cannot conclude that mannose decreases flux into the oxidative and non-oxidative branches of the PPP. Labeling in other pathways have problems. It is very difficult to distinguish the isotopologues because of the colors used and the legend, which has too many possibilities beyond the masses possible. G6P and F6P are not labeled downstream of 13C-glucose (even in low mannose conditions), but are labeled starting at the F1,6P stage, which is highly unlikely since downstream metabolite labeling cannot exceed upstream metabolite labeling. The tracing also does not provide insight into how mannose is impairing nucleotide synthesis – PRPP labeling is similar but IMP levels (but maybe not fraction labeling, it's unclear) dramatically decrease. Labeling with additional tracers (e.g. serine, glutamine, mannose) are really needed to parse where the defects in nucleotide synthesis are occurring, as suggested in the prior round of review.

*Reviewer #2 (Recommendations for the authors):*

The authors have somewhat addressed previous concerns. They evaluated additional cancer cell lines with high or low MPI expression, and the results are consistent with their overall conclusions. The authors have now shown that glutamine depletion reduces viability. However, it is surprising that the high mannose-challenged cells seem more resistant to glutamine starvation than unchallenged cells, given that mannose challenge severely blocks glucose flux into the TCA cycle. This needs to be resolved. Stable isotope tracing studies have not been performed in a satisfactory manner and do not support the manuscript's conclusions. The authors need to properly repeat tracing studies using 1,2-13C-glucose to determine glucose flux into oxidative vs. non-oxidative PPP as well as 13C-gluamine and 13C-mannose. Statistical analysis should be provided for Figure 6. Finally, the authors report that OCR is increased in mannose challenged cells, but there is no significant change in the NAD/NADH ratio. This should be resolved.

---

## [Author Response]

Essential revisions:1) The authors should extend their observations to a larger panel of cell lines, including MPI high and low cell lines.

Thank you very much for this insightful suggestion. With additional experiments carried out, we were able to highlight the phenotypic differences in mannose-caused genomic instability between MPI-KO and MPI^low^ cancer cells.

We extended our observations regarding mannose-caused genomic instability to three human cancer cell lines expressing varying levels of MPI (new Figure 4—figure supplement 3A). These cell lines include one mannose-resistant MPI^high^ cancer cell line (human ovarian cancer SK-OV-3 cells) and two mannose-sensitive MPI^low^ cancer cell lines (human pancreatic cancer KP-4 cells and human non-small-cell lung cancer A549 cells), which were previously characterized by Gonzalez *et al.* (Gonzalez et al., 2018). Consistent with the previous study, we found that mannose challenge (25 mM) significantly suppressed the proliferation of KP-4 and A549 cells, but not that of SK-OV-3 cells (new Figure 4—figure supplement 3B). Having confirmed mannose sensitivity in these cell lines, we analyzed the effects of mannose on the enhancement of cisplatin-induced genomic instability as the expression levels of γH2AX (new Figure 4—figure supplement 3C-F), revealing little enhancement of cisplatin-induced γH2AX by mannose challenge in all three cell lines. This indicates that mannose-caused genomic instability is a unique phenotype observed in the absence of MPI. According to these new findings, we have revised the text in the Result section (from page 12, lines 19-22) and the Discussion section (page 20, lines 4-8).

2) The authors should perform stable isotope tracing experiments to examine the activity of key metabolic pathways (e.g. the pentose phosphate pathway) instead of inferring activity from metabolite pool sizes.

We appreciate the reviewers for raising this important issue. As requested by reviewers 1 and 2, we performed stable isotope labeling experiments using [^13^C_6_]-glucose to examine the activity of key metabolic pathways in MPI-KO HT1080 cells. Reviewer 1 suggested the use of [^13^C_1,2_]-glucose for tracing the pentose phosphate pathway (PPP) flux, but we decided to use [^13^C_6_]-glucose to unambiguously identify the labeled metabolites in both PPP and the downstream nucleotide metabolic pathways. Reviewer 2 suggested the use of stable isotope-labeled mannose and glutamine to strengthen our conclusion on mannose-mediated metabolic changes. However, we focused on the metabolic flux of [^13^C_6_]-glucose in this study because glucose metabolism is a primary target of mannose (Gonzalez et al., 2018). We will further dissect metabolic deficiency in honeybee syndrome by using [^13^C]-mannose and [^13^C,^15^N]-glutamine in a future study.

We analyzed the metabolic flux of [^13^C_6_]-glucose into glycolysis, the tricarboxylic acid (TCA) cycle, the mannose biosynthetic pathway (MBP), PPP and the purine and pyrimidine metabolic pathways. The MPI-KO HT1080 cells were preconditioned under mannose-challenged or unchallenged conditions before [^13^C_6_]-glucose labeling. In mannose-unchallenged cells, [^13^C_6_]-glucose was efficiently metabolized to [^13^C_3_]-lactate and [^13^C_3_]-pyruvate (Figure 6—figure supplement 1 and Data S4), which progressively fueled the TCA cycle over the period of metabolic labeling (Figure 6—figure supplement 2 and Data S4). In contrast, mannose challenge caused an incomplete, but substantial reduction in the flux of [^13^C_6_]-glucose into glycolysis (Figure 6—figure supplement 1 and Data S4), and this glycolytic deficiency reduced the contribution of glucose to the TCA cycle (Figure 6—figure supplement 2 and Data S4), as reported previously (Gonzalez et al., 2018). This result further suggested the importance of glutamine in fueling the TCA cycle in honeybee syndrome.

In MBP, GDP-mannose is produced from GTP and mannose-1-phosphate. In both mannose-challenged and unchallenged cells, GDP-mannose was progressively labeled with [^13^C_5_], which was most likely derived from [^13^C_5_]-ribose (Figure 6—figure supplement 3 and Data S4). Interestingly, we found a significant increase in the steady-state pool of GDP-mannose in mannose-challenged cells, while this increase was not proportional to the increase in the pools of mannose-6-phosphate and mannose-1-phosphate (as the sum with glucose-1-phosphate) (Figure 6—figure supplement 3 and Data S4), implying that the formation of GDP-mannose is a rate-limiting step in MBP.

In PPP, mannose challenge reduced the flux of [^13^C_6_]-glucose to both oxidative and non-oxidative arms in MPI-KO HT1080 cells (Figure 6—figure supplement 4 and Data S4), as reported previously in MPI^low^ cancer cells (Gonzalez et al., 2018). Unexpectedly, however, the levels of [^13^C_5_]-5-phosphoribosyl-1-diphosphate (PRPP), which is synthesized from ribose-5-phosphate, were almost comparable between mannose-challenged and unchallenged cells (Figure 6 and Data S4). More surprisingly, mannose-challenged cells showed very little flux of [^13^C_5_]-PRPP into inosine-5’-phosphate and xanthosine-5’-phosphate in purine metabolism (Figure 6 and Data S4) and orotidine-5’-phosphate in pyrimidine metabolism (Figure 6—figure supplement 5 and Data S4), accompanying nearly the complete loss of newly synthesized pools of (deoxy)ribonucleotides (Figure 6 and Figure 6—figure supplement 5 and Data S4). In accordance with these new findings, we have revised the text in the Result section (from page 15, line 19 to page 17, line 7).

3) The authors should further develop their metabolic characterization of the consequences of the inability to metabolize mannose, including the effects on the TCA cycle and oxidative phosphorylation and dNTP depletion as requested by reviewers #1 and #2.

This request is closely related to the issue raised in Essential revision 2. With additional experiments carried out, we have better characterized the impact of honeybee syndrome on the TCA cycle, oxidative phosphorylation (OXPHOS) and dNTP depletion as follows.

The impact of honeybee syndrome on the TCA cycle and OXPHOS

Our stable isotope labeling experiments clearly demonstrate that mannose challenge substantially reduces the contribution of glucose to fueling the TCA cycle in MPI-KO HT1080 cells (new Figure 6—figure supplement 2). However, the intracellular NAD^+^/NADH ratio remained relatively constant between mannose-challenged and unchallenged cells (new Figure 5—figure supplement 2C, D). Glutaminolysis can contribute to the TCA cycle and thus OXPHOS in cancer cells (Yang et al., 2017). As expected, glutamine depletion greatly reduced the cell viability of both mannose-challenged and mannose-unchallenged cells (new Figure 5—figure supplement 2E, F), suggesting an important role of glutamine in fueling the TCA cycle and OXPHOS in honeybee syndrome. In accordance with these new findings, we have revised the text in the Result section (page 15, lines 6-11).

Although we could not show a clear correlation between the activation states of OXPHOS and NAD^+^/NADH ratio in honeybee syndrome, our seahorse experiments demonstrated that honeybee syndrome increases OCR (Figure 5A, B), which could arise from compensatory activation of glutaminolysis for decreased glucose contribution to the TCA cycle. We will further investigate the impact of honeybee syndrome on mitochondrial functions in a future study. Irrespective of the mechanisms involved, we believe that this limitation will not affect the conclusion of this study, and hope that the reviewers will agree with our view.

The impact of honeybee syndrome on dNTP depletion

The pool size of dNTPs is tightly regulated in order to meet and not to exceed cellular demands by cell cycle-dependent and DNA damage-dependent expression of enzymes involved in the biosynthesis and degradation of dNTPs (Franzolin et al., 2013), as well as their allosteric regulation (Aye et al., 2015; Ji et al., 2013). This tight regulation of dNTP supply is vital to prevent cells from suffering the dNTP pool imbalance, which causes genomic instability (Kumar et al., 2010; Pajalunga et al., 2017). Our proteomic analysis revealed that mannose challenge rapidly increases the expression of RRM2B (Data S1), which is a rate-limiting and TP53-inducible subunit of RNR (Hakansson et al., 2006; Tanaka et al., 2000), indicating that the equilibrium of dNTP metabolism shifted toward biosynthesis at the protein level. However, mannose challenge caused substantial reductions in the pool of ATP, which is a key allosteric regulator for the catalysis of RNR to occur (Brignole et al., 2018). Moreover, mannose challenge impaired the early steps of purine and pyrimidine metabolism, which in turn limited the de novo pool of ribonucleoside diphosphates essential for RNR reaction. Little metabolic flux of PRPP into purine and pyrimidine metabolism in honeybee syndrome can explain why the pool of newly synthesized PRPP was preserved even in the absence of maximal metabolic activity of PPP. A recent in-depth metabolomic study revealed the mannose-mediated downregulation of metabolic activity of glycolysis and PPP in MPI^low^ cancer cells (Gonzalez et al., 2018), although its impact on the downstream nucleotide metabolism remains to be explored. Fueling the salvage pathway of purine and pyrimidine metabolism with hypoxanthine and thymidine (new Figure 6—figure supplement 6A) or with deoxyribonucleoside mixture (new Figure 6—figure supplement 6B) was not sufficient to improve the defects in proliferation of mannose-challenged MPI-KO HT1080 cells, suggesting the critical role of de novo nucleotide biosynthesis in supporting cell proliferation in this cell model. Notably, although dNTPs are essential for DNA repair, the chemosensitizing effect of mannose was rather modest when comparing it with that of deficiency in genes for DNA repair processes (*e.g.*, *BRCA2*, *FANCA* and *FANCD2*) (Bruno et al., 2017; Sakai et al., 2008). This apparent discrepancy most likely arises from the fact that the suppression of de novo nucleotide biosynthesis by honeybee syndrome is incomplete. Taken together, our findings indicate that the insufficient metabolic activity of purine and pyrimidine metabolism in honeybee syndrome causes the loss of dNTPs. Accordingly, we have revised the text in the Discussion section (from page 20, line 17 to page 21, line 22).

4) The authors should demonstrate causality for dNTP depletion in the reduction of proliferation and cell cycle perturbation.

We fully agree with the reviewer’s comment. As discussed in our response to Essential revision 3, fueling the salvage pathway of purine and pyrimidine metabolism with hypoxanthine and thymidine (new Figure 6—figure supplement 6A) or with deoxyribonucleoside (dN) mixture (new Figure 6—figure supplement 6B) was not sufficient to improve the defects in proliferation of mannose-challenged cells, suggesting the critical role of de novo nucleotide biosynthesis in supporting cell proliferation in this cell model. Consistent with this, the supplementation of mannose-challenged or unchallenged MPI-KO HT1080 cells with dN mixture could not improve the incorporation of 5-bromo-2’-deoxyuridine (BrdU) (Author response image 1) and chemosensitivity (Figure R1B, C), except for a slight improvement of the chemosensitivity by dN supplementation in unchallenged cells (Author response image 1). Although we found that honeybee syndrome reduces the biosynthetic capacity of nucleotides, we do not currently understand the precise molecular mechanism behind this metabolic deficiency, making it difficult for us to further investigate the causality of dNTP depletion in the anti-cancer activity of mannose.

**Author response image 1. sa2fig1:** The effects of deoxyribonucleoside supplementation on cell proliferation, DNA synthesis and chemosensitivity in MPI-KO HT1080 cells. (**A**) The percentage of MPI-KO HT1080 (#3) cells that were actively incorporating 2-bromo-5-deoxyuridine (BrdU^+^). Before 1-h pulse labelling with BrdU, the cells were cultured for 24 h in culture medium supplemented with (+) or without (-) a mixture of dNs in the presence of mannose (Man) at the indicated concentrations. (**B and C**) Cell viability in the MPI-KO HT1080 (#3) cells that were precondition for 24 h with (+) or without (-) dN mixture in the presence of mannose at 50 µM (**B**) or 5 mM (**C**), followed by the incubation with cisplatin for additional 24 h under the same culture conditions. Data represent the mean ± SD, n=3 independent experiments. **p<0.01, and ***p<0.001, NS, not significant, Welch’s t-test (A), two-way ANOVA with post hoc Bonferroni’s (**B and C**).

For this reason, we toned down our original title “Metabolic clogging of mannose triggers genomic instability via dNTP loss in human cancer cells” to the new title “Metabolic clogging of mannose triggers dNTP loss and genomic instability in human cancer cells,” and described the limitation of our study in the Discussion section (page 22, lines 14-16) as follows: “Elucidation of the precise molecular mechanism underlying dNTP loss in honeybee syndrome will be necessary to unambiguously identify the direct cause(s) of the chemosensitizing effects of mannose”. However, our findings on the metabolic flux of glucose (new Figure 6) and hydroxyurea treatment (now Figure 7) strongly support our conclusion that dNTP loss is an important mechanism for anti-cancer activity of mannose in our cell models. We hope that the reviewers will agree with our view.

5) The authors should improve data presentation, including showing individual data points and improving the description of the variables as suggested by the reviewers.

We included individual data points where appropriate. We decided not to do so in new Figure 6 and Figure 6—figure supplements 1-5, due to the complexity of the metabolomic data. Instead, we provided the raw data as new Data S4.

We apologize for the poor descriptions on the variables in flow cytometry analysis. We have revised the text in the Results and Figure Legends sections to improve the descriptions on the variables as follows.

– Results section (page 9, lines 2-6)

– Results section (page 10, lines 13-20)

– Results section (page 12, line 22 to page 13, line 5)

– Figure Legends for Figure 2I (DNA replication: page 49, lines 1-3)

– Figure Legends for Figure 3E, 4F and 7C (Chromatin flow cytometry: page 51, lines 11-13; page 53, lines 12-15; page 59, lines 13-14)

– Figure Legends for Figure 4B and 7E (Dormant origin assay: page 53, lines 5-8; page 59, lines 18-21)

6) When giving context to their work, the authors should more thoroughly discuss prior work on cancer cells (not just honeybees), including very in-depth metabolic phenotyping of mannose phosphate isomerase low and high cells that was recently reported.

We fully agree with the reviewer’s comment. We have improved the text by more thoroughly discussing our findings on honeybee syndrome and what is already known about the mannose-mediated metabolic changes in cancer cells.

– Results section (page 12, lines 19-22)

– Results section (page 16, lines 7-11)

– Results section (page 16, lines 21-23)

– Discussion section (page 19, lines 18-20)

– Discussion section (page 21, lines 8-11)

This revision has highlighted a phenotypic similarity between MPI-KO and MPI^low^ cancer cells in mannose-mediated metabolic changes in glycolysis, the TCA cycle and PPP.

Reviewer #1 (Recommendations for the authors):While the manuscript is well-written and advances our understanding of mannose toxicity based on new findings that dNTP depletion occurs, several concerns remain that could be addressed by the authors:1. The metabolic findings from Gonzalez et al. are insufficiently discussed in the text. Their study already established, using sophisticated stable isotope tracing approaches, that mannose decreases the contribution of glucose to G6P and lactate, and results in lower levels of glycolytic and TCA cycle intermediates, and the PPP intermediate ribose-5P. Although the authors cite this study in the context of suppression of proliferation and increased chemotherapeutic efficacy, what is already known about mannose metabolism in mammalian cells is understated in the manuscript and the focus on honeybees is overemphasized.

We fully agree with the reviewer’s comment. We have addressed this issue in our response to Essential revision 6.

2. Gonzalez et al., found that mannose decreased ribose-5P but in this study no difference in ribose-5P was seen upon mannose treatment. Total measurement of PPP metabolites is not indicative of pathway flux and the authors should instead use 1,2-13C-glucose tracing to determine whether mannose influences PPP flux. Moreover, U-13C-glucose can be used to look at dNTP labeling.

We have addressed this issue in our response to Essential revision 2.

3. The mechanistic connection between mannose metabolism and dNTP depletion is not established. If there is no effect of mannose on the PPP as claimed, how does mannose induce dNTP depletion to induce replication stress?

We have addressed this issue in our response to Essential revisions 2 and 3.

4. It is unclear whether dNTP depletion is a cause or consequence of reduced proliferation and cell cycle perturbation. Direct evidence is lacking. Can nucleosides rescue mannose toxicity, cell cycle progression, chemosensitivity and DNA synthesis?

We have addressed this issue in our response to Essential revision 4.

Reviewer #2 (Recommendations for the authors):1. Identification of cancer cell lines that may be susceptible to high mannose-induced toxicity, such as those with endogenous low expression levels of MPI, and demonstration of the same effects in those cell lines would strengthen the study.

We have addressed this issue in our response to Essential revision 1.

2. Utilization of labeled isotope tracers such as 13C-glucose, 13C-mannose, or 13C-glutamine to trace metabolic flux would provide confidence in the conclusions about the changes to the metabolic landscape. Pool sizes decreased for both lower glycolysis and TCA cycle intermediates; this was interpreted as decreased activity for glycolysis but increased activity for the TCA cycle. How is the TCA cycle fueled if lower glycolytic intermediates are not feeding the TCA cycle? Is excess glutamine fueling the TCA cycle?

We have addressed this issue in our response to Essential revisions 2 and 3.

Why are nucleoside monophosphates not depleted?

Despite our extensive metabolic characterization of honeybee syndrome by using [^13^C_6_]-glucose, the mechanism by which honeybee syndrome hardly depletes the steady state pool of ribonucleoside monophosphates remained unknown. One plausible explanation for this phenomenon is that honeybee syndrome severely reduces the biosynthesis of nucleoside di/triphosphates from both newly synthesized and pre-existing pools of ribonucleoside monophosphates. Irrespective of the mechanisms involved, we believe that this limitation will not affect our conclusion, and hope that the reviewers will agree with our view.

3. Evaluating the NAD/NADH ratio between mannose challenged and unchallenged cells would also strengthen the claims about increased oxidative phosphorylation.

We addressed this issue in our response to Essential revision 3.

4. For Seahorse experiments, data following oligomycin, rotenone, and antimycin A addition should be included.

We apologize for our poor presentation of the data regarding seahorse experiments. We have included the data following the administration of oligomycin A and a mixture of rotenone and antimycin A (Figure 5A, B) and revised the Results as follows (from page 13, line 22 to page 14, line 4):

“Mannose challenge increased OCR in response to a steep drop of ECAR (Figure 5A, B), indicating that mannose challenge rapidly inhibited glycolysis, which in turn activated OXPHOS. The remaining ECAR further decreased after oligomycin A treatment in mannose-challenged cells, while the same treatment increased ECAR in mannose-unchallenged cells (Figure 5A), indicating that mannose challenge ablates glycolytic capacity, which is required to buffer the defects in OXPHOS.”

5. Figure 1G: would removal of mannose further reduce hexose-6-phosphate levels?

Exogenous mannose is required for the production of mannose-6-phosphate and the downstream metabolites (*i.e.*, mannose-1-phosphate and GDP-mannose) in the absence of MPI, and therefore the removal of mannose from culture medium should reduce all of these mannose-related metabolites and impair protein N-glycosylation (Harada et al., 2013; Harada et al., 2021). Although we did not directly measure the levels of mannose-6-phosphate, the fact that mannose-starved MPI-KO HT1080 cells showed poor protein N-glycosylation (Figure 1H) supports our idea that mannose starvation reduces hexose-6-phosphate levels.

6. When cell growth is restored after re-introduction of physiologic mannose as shown in Figure 2G, will these cells revert back to a normal cell cycle progression as shown in Figure 2B, or do they retain aberrant cell cycle progression patterns?

Thank you very much for this insightful suggestion. We employed Fucci(CA) to compare cell cycle progression between the MPI-KO HT1080 cells that were (i) maintained under mannose-unchallenged conditions or (ii) recovered from mannose challenge as in Figure 2G. The results indicate that the recovered cells show cell cycle progression that is indistinguishable from that of the cells maintained under mannose-unchallenged conditions (new Figure 2—figure supplement 2A–D). In accordance with this new finding, we have revised the text in the Result section (page 8, lines 20-24).

7. Figure 2I and J do not show BRdU incorporation into MPI-KO HT1080 cells under 50 μM mannose on day 6. Similarly, Figure 2H does not include p21 and p27 expression for day 6 in the 50 μM mannose condition. Please justify.

We could not analyze the 6-day culture of mannose-unchallenged cells due to their overgrowth.

8. Inclusion of individual data points for each replicate within bar graphs would improve the reader's ability to interpret results.

We have addressed this issue in our response to Essential revision 5, which, we believe, significantly improved the reader’s ability to interpret the results.

Reviewer #3 (Recommendations for the authors):Specific Points:The data in Figure 1. The induction of honeybee syndrome suppresses cell proliferation and increases chemosensitivity. Shows only modest changes in chemosensitivity and the discussion does not adequately express this. The authors should further discuss this.

We thank the reviewer for raising this important issue. Although dNTPs are essential for DNA repair, the chemosensitizing effect of mannose was rather modest when comparing it with that of deficiency in genes for DNA repair processes (*e.g*., *BRCA2*, *FANCA* and *FANCD2*) (Bruno et al., 2017; Sakai et al., 2008). This apparent discrepancy most likely arises from the fact that the suppression of de novo nucleotide biosynthesis in honeybee syndrome is incomplete. We revised the text in the Discussion section accordingly (page 21, lines 16-20).

Figure 3. These results are discussed almost exclusively in the context of DNA replication and cell cycle for the down regulated pathways. What about ribosomes?

We have revised the text in the Discussion section (page 21, line 23 to page 22 line 9) to more thoroughly discuss our proteomic data, including those regarding ribosomes, as follows:

“Our proteomic analysis revealed the functional pathways affected by honeybee syndrome. The downregulation of MCM2-7 proteins in mannose-challenged cells supports our hypothesis that honeybee syndrome impairs dormant origins and causes genomic instability (Ibarra et al., 2008; Kawabata et al., 2011; Shima et al., 2007). Mannose challenge also decreased the expression of several large ribosomal subunit proteins. Ribosome biogenesis stress has been identified as a major mechanism by which oxaliplatin kills cancer cells in a DNA damage response-independent manner (Bruno et al., 2017). Moreover, mannose challenge upregulated proteins involved in ferroptosis and necroptosis. These two programmed cell death mechanisms are considered as an effective approach to overcome chemoresistance (Zhang et al., 2022) or to eradicate apoptosis-resistant cancer cells (Su et al., 2016). In addition to metabolic impacts, honeybee syndrome may also enhance cancer cell vulnerability at the proteomic level.”

We hope that the reviewer will find our revision sufficient for discussing our findings on ribosomes.

The data are clear but it is hard to assess the variability in some of the assays since there is scant description of the variables particularly the cell sorting experiments to establish DNA replication and cell cycle.

We apologize for our poor descriptions on the variables in cell sorting experiments. We have addressed this issue in our response to Essential revision 5.

References

Aye, Y., Li, M., Long, M.J., and Weiss, R.S. (2015). Ribonucleotide reductase and cancer: biological mechanisms and targeted therapies. Oncogene *34*, 2011-2021.

Brignole, E.J., Tsai, K.L., Chittuluru, J., Li, H., Aye, Y., Penczek, P.A., Stubbe, J., Drennan, C.L., and Asturias, F. (2018). 3.3-A resolution cryo-EM structure of human ribonucleotide reductase with substrate and allosteric regulators bound. *eLife 7*.

Bruno, P.M., Liu, Y., Park, G.Y., Murai, J., Koch, C.E., Eisen, T.J., Pritchard, J.R., Pommier, Y., Lippard, S.J., and Hemann, M.T. (2017). A subset of platinum-containing chemotherapeutic agents kills cells by inducing ribosome biogenesis stress. Nat Med *23*, 461-471.

Franzolin, E., Pontarin, G., Rampazzo, C., Miazzi, C., Ferraro, P., Palumbo, E., Reichard, P., and Bianchi, V. (2013). The deoxynucleotide triphosphohydrolase SAMHD1 is a major regulator of DNA precursor pools in mammalian cells. Proc Natl Acad Sci U S A *110*, 14272-14277.

Gonzalez, P.S., O'Prey, J., Cardaci, S., Barthet, V.J.A., Sakamaki, J.I., Beaumatin, F., Roseweir, A., Gay, D.M., Mackay, G., Malviya, G.*, et al.* (2018). Mannose impairs tumour growth and enhances chemotherapy. Nature *563*, 719-723.

Hakansson, P., Hofer, A., and Thelander, L. (2006). Regulation of mammalian ribonucleotide reduction and dNTP pools after DNA damage and in resting cells. J Biol Chem *281*, 7834-7841.

Harada, Y., Nakajima, K., Masahara-Negishi, Y., Freeze, H.H., Angata, T., Taniguchi, N., and Suzuki, T. (2013). Metabolically programmed quality control system for dolichol-linked oligosaccharides. Proc Natl Acad Sci U S A *110*, 19366-19371.

Harada, Y., Ohkawa, Y., Maeda, K., and Taniguchi, N. (2021). Glycan quality control in and out of the endoplasmic reticulum of mammalian cells. FEBS J.

Ji, X., Wu, Y., Yan, J., Mehrens, J., Yang, H., DeLucia, M., Hao, C., Gronenborn, A.M., Skowronski, J., Ahn, J.*, et al.* (2013). Mechanism of allosteric activation of SAMHD1 by dGTP. Nat Struct Mol Biol *20*, 1304-1309.

Kumar, D., Viberg, J., Nilsson, A.K., and Chabes, A. (2010). Highly mutagenic and severely imbalanced dNTP pools can escape detection by the S-phase checkpoint. Nucleic Acids Res *38*, 3975-3983.

Pajalunga, D., Franzolin, E., Stevanoni, M., Zribi, S., Passaro, N., Gurtner, A., Donsante, S., Loffredo, D., Losanno, L., Bianchi, V.*, et al.* (2017). A defective dNTP pool hinders DNA replication in cell cycle-reactivated terminally differentiated muscle cells. Cell Death Differ *24*, 774-784.

Sakai, W., Swisher, E.M., Karlan, B.Y., Agarwal, M.K., Higgins, J., Friedman, C., Villegas, E., Jacquemont, C., Farrugia, D.J., Couch, F.J.*, et al.* (2008). Secondary mutations as a mechanism of cisplatin resistance in BRCA2-mutated cancers. Nature *451*, 1116-1120.

Tanaka, H., Arakawa, H., Yamaguchi, T., Shiraishi, K., Fukuda, S., Matsui, K., Takei, Y., and Nakamura, Y. (2000). A ribonucleotide reductase gene involved in a p53-dependent cell-cycle checkpoint for DNA damage. Nature *404*, 42-49.

Yang, L., Venneti, S., and Nagrath, D. (2017). Glutaminolysis: A Hallmark of Cancer Metabolism. Annu Rev Biomed Eng *19*, 163-194.

[Editors' note: further revisions were suggested prior to acceptance, as described below.]

The manuscript has been improved but there are some remaining issues that need to be addressed, as outlined below:The stable isotope tracing experiments have problems that leave this point unaddressed. Specifically:1. It is very difficult to distinguish the isotopologues because of the colors used and the legend, which has too many possibilities beyond the masses possible. The authors should simplify these figures and use colors that are more easily distinguishable.

We apologize for the poor data presentation, which made our data uninterpretable. We have revised the figures by estimating fractional enrichment, by showing the possible isotopologues for a metabolite, and by using colors that are more easily distinguishable (revised Figure 6 and Figure 6—figure supplements 1-3).

2. U-13C6-glucose is not suitable to assay the non-oxidative and oxidative branches of the PPP and thus the authors cannot conclude that mannose decreases flux into this pathway. The authors should use 1,2-C2-glucose tracing to draw these conclusions and also perform labeling with additional tracers (e.g. serine, glutamine, mannose) to parse where the defects in nucleotide synthesis are occurring.

It is technically difficult to accurately estimate the dynamic labeling for PPP intermediates, due to their very rapid saturation as pointed by reviewers and literatures (please see also our response to the Reviewer 1’s comment). For this reason, we have removed our claim for the PPP flux from manuscript.

To substantiate our claim that mannose impairs nucleotide biosynthesis, we re-analyzed data obtained from our U-^13^C_6_-glucose tracing experiments. The data have clearly indicated that mannose causes sever defects in the early stages of purine and pyrimidine metabolism (revised Figure 6 and Figure 6—figure supplements 1-3). We hope that the reviewers will find these revisions sufficient for drawing our conclusion.

3. The authors should address other issues raised in the comments below.

Please see our responses to the reviewers’ comments below.

Reviewer #1 (Recommendations for the authors):The authors have done a good job addressing concerns about (1) extending their observations to a larger panel of MPI high and low cell lines, (2) further develop their characterization of the effects of mannose on the TCA cycle and dNTPs, and (3) examine the causality for dNTP depletion in genomic instability.

Thank you for finding our manuscript being improved.

However, the stable isotope tracing experiments have problems that leave this point unaddressed. The authors chose U-13C6-glucose instead of 1,2-C2-glucose, and this is suitable to assay the TCA cycle. However, it is unsuitable to assay the non-oxidative and oxidative branches of the PPP because these rapidly saturate and labeling in R5P from either branch cannot be distinguished. This is very evident in Figure 6—figure supplement 4, where both 6PG and S7P are completely labeled at all time points and both conditions, with only the total levels different. It is unclear why mannose decreases the fraction labeling in Ru5P + R5P specifically. Therefore, the authors cannot conclude that mannose decreases flux into the oxidative and non-oxidative branches of the PPP.

The reviewer is correct that 1,2-^13^C_2_-glucose is suitable for analyzing oxidative and non-oxidative arms of the PPP. However, as explained in our response to Essential revision 2, it is technically difficult to accurately estimate the dynamic labeling for metabolic intermediates of the PPP, as well as those of glycolysis, due to their very rapid saturation. This has also been evident in many excellent literatures, showing that isotopic steady state of glycolysis can be reached within minutes of ^13^C_6_-glucose administration (Lorkiewicz et al., 2019), and that glycolysis and the PPP often label at similar rates (Jang et al., 2018). Accordingly, we have revised the text by removing our claims for the flux of glycolysis and the PPP, but leaving descriptions for their rapid saturation (page 15, lines 19-22).

Labeling in other pathways have problems. It is very difficult to distinguish the isotopologues because of the colors used and the legend, which has too many possibilities beyond the masses possible.

We have addressed this issue in our response to Essential revision 1.

G6P and F6P are not labeled downstream of 13C-glucose (even in low mannose conditions), but are labeled starting at the F1,6P stage, which is highly unlikely since downstream metabolite labeling cannot exceed upstream metabolite labeling.

The reviewer raised a concern about precursor-product relationships. However, as mentioned above, isotopic steady state of glycolytic intermediates can be reached within minutes after the addition of ^13^C_6_-glucose (Lorkiewicz et al., 2019), making it technically difficult to address this issue in a satisfactory manner.

The tracing also does not provide insight into how mannose is impairing nucleotide synthesis – PRPP labeling is similar but IMP levels (but maybe not fraction labeling, it's unclear) dramatically decrease.

To substantiate our claim that mannose impairs nucleotide biosynthesis, we reanalyzed data obtained from ^13^C_6_-glucose tracing experiments by estimating fractional enrichment of metabolites in a dynamic or sub-dynamic labeling phase. The results clearly indicate that mannose causes sever defects in the early stages of purine and pyrimidine metabolism as follows.

To further elucidate the impacts of mannose challenge on the biosynthesis of dNTPs, we performed [^13^C_6_]-glucose tracer experiments that, unlike steady-state metabolomics, allow to estimate the activity of glucose-related metabolic pathways by analyzing the fractional enrichment of the metabolites in a dynamic labeling phase (Lorkiewicz et al., 2019). The MPI-KO HT1080 cells were preconditioned under mannose-challenged or -unchallenged conditions before [^13^C_6_]-glucose labeling. The ^13^C-labeling of most metabolites detected in glycolysis, the PPP and the TCA cycle (Figure 6—figure supplement 1A) already reached isotopic steady states within 30 min in both mannose-challenged and unchallenged cells (Figure 6—figure supplement 1B), most likely due to their high metabolic activity (Jang et al., 2018; Lorkiewicz et al., 2019). In contrast to these central metabolic pathways, metabolites in purine and pyrimidine metabolism showed a dynamic or sub-dynamic labeling in mannose-challenged cells.

Phosphoribosyl pyrophosphate (PRPP) is an essential ribose donor substrate for the biosynthesis of nucleotides in both purine and pyrimidine metabolism (Figure 6A, B and Figure 6—figure supplement 2A). Fractional enrichment of ^13^C-PRPP (M+5) was saturated at 30 min in unchallenged cells, while it was still in a sub-dynamic labeling phase in mannose-challenged cells (Figure 6C). Despite this slower fractional enrichment, the pool size of ^13^C-PRPP (M+5) in mannose-challenged cells was similar to that in unchallenged cells (Figure 6D), implying that mannose challenge reduced the utilization of PRPP for nucleotide biosynthesis. Consistent with this assumption, we found a marked reduction in both the fractional enrichment and the pool size of ^13^C-purine metabolic intermediates (M+5) in mannose-challenged cells, which included inosine-5’-monophosphate (IMP; Figure 6 E, F), adenosine-5’-monophosphate (AMP; Figure 6 G, H) and guanosine-5’-monophosphate (GMP; Figure 6 I, J).

PRPP is utilized for both de novo and salvage synthesis of purine nucleotides (Figure 6A), and therefore the M+5 fraction of ^13^C-IMP, ^13^C-AMP and ^13^C-GMP can be accounted for the sum of their de novo and salvage pools. In contrast, purine nucleotides with the labeled fractions greater than M+5 can originate from de novo synthesis (Figure 6A). We found a progressive increase in the M+6 fraction of ^13^C-IMP, ^13^C-AMP and ^13^C-GMP in unchallenged cells (Figure 6E, G and I), suggesting that ^13^C-10-formyl-tetrahydrofolate (CHO-THF; M+1), which is produced de novo via the glycolysis-serine biosynthesis-folate cycle (GSF) axis (Figure 6A) (Yang and Vousden, 2016), contributed to the de novo synthesis of purine nucleotides. Although we could not directly detect ^13^C-labeling in serine and CHO-THF in our metabolomic analysis, the fractional enrichment and the pool size of ^13^C-glycine (M+2), which is a signature metabolite produced in coupled with 5,10-methylenetetrahydrofolate via the GSF axis, progressively increased in unchallenged cells (Figure 6K, L). However, the fractional enrichment and the pool size of ^13^C-glycine (M+2) largely decreased in mannose-challenged cells (Figure 6K, L), suggesting that the GSF axis is compromised in these cells. These results may partly explain why the de novo synthesis of purine nucleotides is limited in honeybee syndrome.

In the early stage of pyrimidine metabolism, aspartate (Asp) is transferred to carbamoyl phosphate, giving rise to *N*-carbamoyl aspartate (carbamoyl Asp) (Figure 6—figure supplement 2A). We found a progressive increase in the M+2, M+3 and M+4 fractions of both ^13^C-Asp (Figure 6—figure supplement 2B, C) and ^13^C-carbamoyl Asp (Figure 6—figure supplement 2D, E) at similar labeling rates in unchallenged cells. The ^13^C-Asp (M+2, M+3 and M+4) could originate from [^13^C_6_]-glucose-derived oxaloacetate that is formed in the first, second and third rounds of the TCA cycle (Figure 6—figure supplement 2F), as indicated by the presence of the M+2, M+3 and M+4 fractions of ^13^C-malate (Figure 6—figure supplement 2G, H). However, mannose challenge severely decreased the fractional enrichment of both ^13^C-Asp (M+2, M+3 and M+4) and ^13^C-carbamoyl Asp (M+2, M+3 and M+4) (Figure 6—figure supplement 2B, D). In contrast, the pool size of ^13^C-Asp (M+2) remained relatively unchanged between mannose-challenged and -unchallenged cells (Figure 6—figure supplement 2C), while the pool size of ^13^C-carbamoyl Asp (M+2) greatly decreased in mannose-challenged cells (Figure 6—figure supplement 2E), indicating that mannose challenge reduced the utilization of Asp in pyrimidine metabolism. In the immediate downstream metabolites of carbamoyl Asp, we could detect significant amounts of uridine-5’-monophosphate (UMP), and found that a large majority of ^13^C-UMP formed in unchallenged cells was consisted of the M+5 fraction (Figure 6—figure supplement 2I, J). This fraction was most likely to originate from unlabeled carbamoyl Asp (Figure 6—figure supplement 2K) and ^13^C-PRPP (M+5). Moreover, we identified the M+6, M+7 and M+8 fractions of ^13^C-UMP in unchallenged cells (Figure 6—figure supplement 2I), indicating that both ^13^C-carbamoyl Asp (M+2, M+3 and M+4) and ^13^C-PRPP (M+5) contributed to forming ^13^C-orotidine-5’-monophosphate (M+7, M+8 and M+9), which is decarboxylated to give ^13^C-UMP (M+6, M+7 and M+8). However, mannose-challenged cells showed a substantial reduction of ^13^C-UMP in both the fractional enrichment (M+5, M+6, M+7 and M+8) and the pool size (M+5) (Figure 6—figure supplement 2I, J), clearly indicating that mannose challenge impaired the biosynthesis of pyrimidine nucleotides. As expected, dNTPs showed little ^13^C enrichment (dATP and dGTP, Figure 6—figure supplement 3A−E) and a very low ^13^C enrichment (dTTP and dCTP, Figure 6—figure supplement 3F−J) in mannose-challenged cells as compared with those in unchallenged cells. Taken all these findings together, mannose challenge impairs both purine and pyrimidine metabolism at the early stage, thereby potentially limiting the de novo synthesis of dNTPs.

Accordingly, we revised the Results section (from page 15, line 18 to page 18, line 17) and the Discussion section (page 22, lines 4-8 and page 22, lines 12-13).

Labeling with additional tracers (e.g. serine, glutamine, mannose) are really needed to parse where the defects in nucleotide synthesis are occurring, as suggested in the prior round of review.

The results obtained from our [^13^C_6_]-glucose tracing experiments have now indicated that mannose challenge impairs the early stages of the de novo synthesis of purine and pyrimidine nucleotides. For this reason, we believe that the additional tracer experiments are not essential to draw our conclusion.

Reviewer #2 (Recommendations for the authors):The authors have somewhat addressed previous concerns. They evaluated additional cancer cell lines with high or low MPI expression, and the results are consistent with their overall conclusions. The authors have now shown that glutamine depletion reduces viability. However, it is surprising that the high mannose-challenged cells seem more resistant to glutamine starvation than unchallenged cells, given that mannose challenge severely blocks glucose flux into the TCA cycle. This needs to be resolved.

It is beyond the scope for this study to further demonstrate how mannose-challenged cells can be more tolerated with glutamine depletion than unchallenged cells. For this reason, we have removed the uninterpretable data (Figure 5—figure supplements 2E, F) and the related statements from manuscript.

Stable isotope tracing studies have not been performed in a satisfactory manner and do not support the manuscript's conclusions. The authors need to properly repeat tracing studies using 1,2-13C-glucose to determine glucose flux into oxidative vs. non-oxidative PPP as well as 13C-gluamine and 13C-mannose.

Please see our responses to Essential revision 2 and the Reviewer 1’s comment.

Statistical analysis should be provided for Figure 6.

We have performed statistical analysis on data presented in revised Figure 6 and revised Figure 6—figure supplements 1-3.

Finally, the authors report that OCR is increased in mannose challenged cells, but there is no significant change in the NAD/NADH ratio. This should be resolved.

It is beyond the scope for this study to further demonstrate why the NAD/NADH ratio did not change in mannose-challenged cells. For this reason, we have removed the uninterpretable data (Figure 5—figure supplements 2C, D) and the related descriptions from manuscript. We feel appropriate to tone down our claim for the activation of OXPHOS by mannose challenge, as a slight increase in OCR is the only evidence supporting this hypothesis. We have revised the text accordingly (page 13, line 22 to page 14, line 1).

References

Jang, C., Chen, L., and Rabinowitz, J.D. (2018). Metabolomics and Isotope Tracing. Cell *173*, 822-837.

Lorkiewicz, P.K., Gibb, A.A., Rood, B.R., He, L., Zheng, Y., Clem, B.F., Zhang, X., and Hill, B.G. (2019). Integration of flux measurements and pharmacological controls to optimize stable isotope-resolved metabolomics workflows and interpretation. Sci Rep *9*, 13705.

Yang, M., and Vousden, K.H. (2016). Serine and one-carbon metabolism in cancer. Nat Rev Cancer *16*, 650-662.